# Is Fixing Schema Graphs Necessary?
# Full-Resolution Graph Structure Learning for Relational Deep Learning

Yi Huang[1]  Qingyun Sun[1][†]  Jia Li[1]  Xingcheng Fu[2]  Jianxin Li[1]

## Abstract

Relational prediction tasks are fundamental in many real-world applications, where data are naturally stored in relational databases (RDBs). Relational Deep Learning (RDL) addresses this problem by modeling RDBs as graphs and applying graph neural networks (GNNs) for end-to-end learning. However, the full-resolution property is commonly adopted as a design principle in graph construction for RDBs to preserve relational semantics, which leads most existing methods to rely on *fixed* graph structures. In this paper, we propose **FROG**, a *Full-Resolution and Optimizable Graph Structure Learning* framework for RDL that formulates relational structure learning as a learnable table role modeling problem, allowing tables to contribute as nodes and edges in message passing. We further design role-driven message passing mechanisms to capture relational semantics, enabling joint optimization of graph structure and GNN representations. To ensure semantic consistency, we introduce functional dependency constraints that regularize representations across table and entity levels. Extensive experiments demonstrate that our method outperforms existing approaches and reveal how table roles impact downstream tasks, offering new insights into graph construction for RDL[1].

## 1. Introduction

Many real-world applications organize data as relational databases (RDBs) (Harrington, 2016), where multiple en-

[1]SKLCCSE, School of Computer Science and Engineering, Beihang University, Beijing, China [2]Key Lab of Education Blockchain and Intelligent Technology, Ministry of Education, Guangxi Normal University, Guilin, China. Correspondence to: Qingyun Sun <sunqy@buaa.edu.cn>.

*Proceedings of the 43rd International Conference on Machine Learning*, Seoul, South Korea. PMLR 306, 2026. Copyright 2026 by the author(s).

[1]Code is available at: https://github.com/RingBDStack/FROG

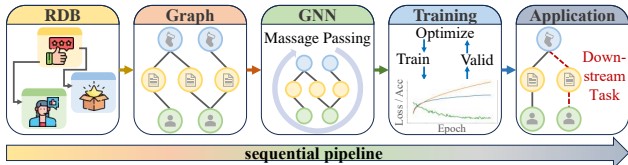

*Figure 1.* Pipeline of Typical RDL Method

tities are connected through explicit relations to describe complex systems. Because of this structured organization, such relational data is large in scale and rich in structural information, making it central to practical decision-making. Accordingly, a wide range of predictive tasks are defined on relational databases (Robinson et al., 2024; Wang et al., 2024), such as forecasting users' future reviewing behavior based on `customer`, `product`, and `review` tables.

Traditional solutions typically flatten relational data into a single table, followed by manual feature engineering using tabular models (Chen & Guestrin, 2016; Kaggle, 2022). While effective in simple settings, this approach is labor-intensive and often discards relational structure, resulting in weaker predictive signals. These limitations have motivated Relational Deep Learning (RDL) (Fey et al., 2024), which aims to perform end-to-end learning directly over RDBs by jointly modeling the relations between entities and learning their dependencies across multiple tables. Most existing RDL methods follow a sequential pipeline: a Relational Entity Graph (REG) is first constructed based on a *fixed* Schema Graph, and a graph neural network (GNN) (Gilmer et al., 2017; Wu et al., 2020) is then trained on top of this for end-to-end learning, as illustrated in Figure 1. However, this raises a natural question:

*Is Fixing Schema Graphs Necessary?* — Maybe Not.

The way a graph is constructed directly constrains the information accessible during message passing (Zhong et al., 2023; Zhuo et al., 2023; Yang et al., 2025). As a result, relations that are implicit, high-order, or task-specific may be difficult to expose, limiting the effectiveness of downstream GNN learning (as shown in Figure 2). Consequently, the graph should *not merely serve as a representation of the RDB, but should evolve together with GNNs*.

However, the graph is not the primary object of interest;

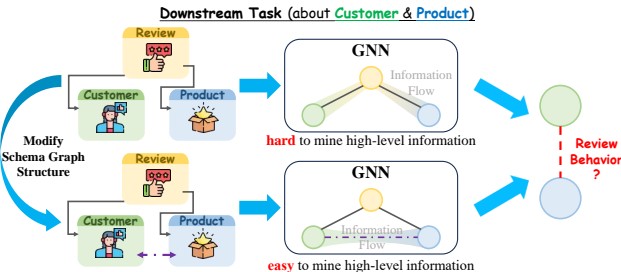

*Figure 2.* Illustration of the impact of graph structure

rather, the RDB itself constitutes the core input, and the graph serves only as an alternative representation, which is expected to preserve two essential relational properties: *full-resolution* and *functional dependencies*. ① The *full-resolution* property was first introduced by Fey et al. (2024), which requires each entity and foreign key link in the RDB to be faithfully encoded in the graph based on the schema, ensuring that *no relational information is lost*. Under this constraint, traditional graph structure learning (GSL) approaches (Zhu et al., 2021) that rely on pruning or adding edges are incompatible in RDL, as such operations compromise the full-resolution property (Sec. 3.3). ② Moreover, maintaining the integrity of relational data requires respecting key constraints, most notably *Functional Dependencies*, where one entity uniquely determines another across tables (e.g., $review_1 \rightarrow user_1$ but $review_1 \nrightarrow user_2$). Together, this calls for *rethinking graph structure optimization problem in RDL from a new perspective*.

In this context, RDB2G (Choi et al., 2025) takes an early step toward graph modeling for RDL. It models relational tables as edges and studies how different schema graph constructions affect downstream GNN performance by exhaustively enumerating candidate graphs. While insightful, RDB2G focuses on discrete *role selection* and empirical comparison, without addressing *how table roles (or schema graph) can be optimized* in an end-to-end manner.

Therefore, developing a *full-resolution* and *optimizable* GSL method that maintains *functional dependencies* to capture relational structures remains an important open problem. Motivated by RDB2G, We follow the idea of modeling relational tables not only as nodes, but also as edges. However, this approach still faces the following challenges:

- How to guarantee that the graph construction satisfies the full-resolution property in Def. 3.2? (▷ Sec. 4.1)
- How to optimize table roles in a learnable manner, rather than discrete role selection? (▷ Sec. 4.2)
- How to model relation-driven information flow across tables with different roles? (▷ Sec. 4.3)
- How to enable the model to maintain the "functional dependency" during learning processes? (▷ Sec. 4.4)

In this work, we propose **FROG**, a *Full-Resolution and*

*Optimizable Graph Structure Learning* framework. To the best of our knowledge, it is the first to introduce GSL into RDL. We first investigate the different effects of modeling tables as nodes or edges while satisfying the full-resolution property, and demonstrate that modeling table-as-edge better captures long-range dependencies via mutual information. Then, we design relation-driven message-passing paradigms to learn diverse relational information and enable structure optimization. Finally, we introduce functional dependency loss at both table and entity levels to encourage that FD constraints in RDBs are maintained. Our contributions are:

- We conduct an in-depth analysis of the full-resolution property of RDBs from an information-theoretic perspective and propose a graph structure learning framework FROG that preserves full-resolution.
- We optimize table roles by designing relation-driven message passing paradigms, enabling adaptive node-edge role learning for Schema Graph construction.
- We design the functional dependency loss to constrain the embedding space, enabling the model to respect functional dependency constraints inherent in RDBs.
- Extensive experiments demonstrate the effectiveness of our approach and systematically analyze the impact of table-as-node and table-as-edge representations.

## 2. Related Work

### 2.1. Relational Deep Learning

Relational Deep Learning (RDL), aims to solve the prediction problem on relational databases in an end-to-end manner using GNNs (Fey et al., 2024; Dwivedi et al., 2025b). Robinson et al. (2024), Wang et al. (2024) and Choi et al. (2025) have built benchmarks on RDB, establishing a framework for RDL learning. Based on this, RelGNN (Chen et al., 2025) achieves information transmission of complex relationships on RDB by introducing atomic paths. RelGT (Dwivedi et al., 2025a) and RGP (Lachi et al., 2025) extract features based on graph transformer, while Truong et al. (2025) proposed a pre-training framework and Wang et al. (2025) proposed the foundation model "Griffin" on RDB. However, existing methods assume a fixed graph structure once constructed, which limits the model-graph coupling and may reduce the model's performance.

### 2.2. Graph Structure Learning

Graph Structure Learning (GSL) (Jin et al., 2020; Sun et al., 2022; Yuan et al., 2025) aims to optimize both the graph structure and the model simultaneously. It enables the model to dynamically discover the most informative connections, rather than relying solely on a predefined graph. By mining the relationships between node pairs, the edges in the graph

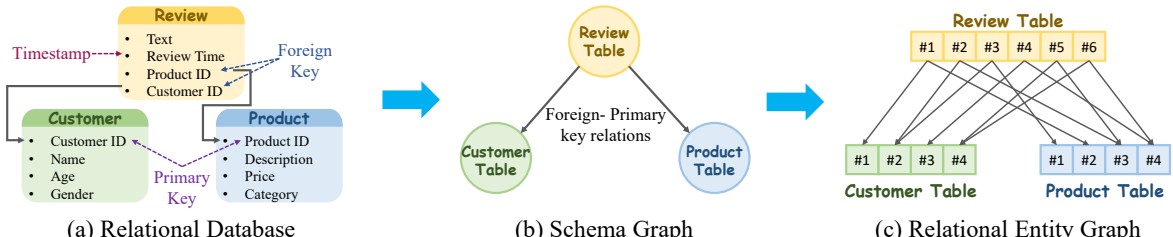

*Figure 3.* Overview of the core concepts of RDL

can be added, deleted, or modified. This is mostly achieved by modifying the adjacency matrix or Laplacian matrix of the graph (Zhu et al., 2021; Li et al., 2018). GSL makes graphs more compatible with models and downstream tasks, achieving collaborative updates of graphs and models. However, current GSL methods are inherently incompatible with RDL mapping: ① The model learns connections between node pairs rather than table relations, and may violate the functional dependencies in Def. 3.1. ② Graph structure changes may *break full-resolution with respect to the RDB*, violating a key property required for modeling RDBs.

## 3. Background and Problem Formulation

### 3.1. Relational Database

The Relational Database (RDB) is a widely used data storage model which can be formally represented as $(\mathcal{T}, \mathcal{R})$. Here, $\mathcal{T} = \{T_1, \dots, T_n\}$ represents a set of tables in the RDBs and $\mathcal{R} \subset \mathcal{T} \times \mathcal{T}$ is the set of relations between these tables. Each table $T \in \mathcal{T}$ is composed of a series of entities (or rows) $v \in T$, and each entity $v$ carries the attribute $x_v$ and an optional timestamp $t_v$. The structural connectivity of RDB is maintained through Foreign-Primary Keys (FPKs): each entity has a unique primary key for identification, and foreign keys link entities across tables.

Functional Dependency (FD) (Armstrong, 1974) is a fundamental concept in relational databases, capturing intrinsic constraints among attributes that govern data consistency and normalization in relational schemas:

**Definition 3.1** (Functional Dependency). In RDBs, a functional dependency is a constraint between two sets of attributes in a relation. Given two attribute sets $X$ and $Y$, a functional dependency, denoted as $X \rightarrow Y$, holds if and only if for each unique value of $X$, there exists exactly one corresponding value of $Y$.

For example, in `Review` table with attributes (review_id, customer_id, product_id, rating), the FD: review_id $\rightarrow$ {customer_id, product_id, rating} holds, since each review uniquely identifies the customer who wrote it, the product being reviewed and the assigned rating.

### 3.2. Relational Deep Learning

Relational Deep Learning (RDL) aims to achieve end-to-end predictive modeling of RDBs (Fey et al., 2024), thereby avoiding time-consuming and complex manual feature engineering. To capture relational structure, a RDB is modeled as a "*Schema Graph*", where nodes represent tables and directed edges denote foreign-primary key relations. This representation not only describes data organization, but more importantly, it lays the foundation for the transformation of RDB into a temporal heterogeneous "*Relational Entity Graph*" (REG), $\mathcal{G}$. The node attributes of REG contain the feature vector $x_v \in X$ of the entity itself, and the structures are determined by the schema graph. An intuitive diagram of a RDB and its schema graph is shown in Figure 3.

The REG enables GNNs to propagate information across multi-table associations via message passing. In particular, the timestamp $t_v$ in RDB plays a crucial role in REG, because real data often depends on the time relationship. Beyond serving as node features, they enforce temporal causality in time-series prediction, ensuring that message passing at prediction time $t_{predict}$ only incorporates information from $t_v \leq t_{predict}$ and prevents future information leakage. This can be effectively restricted via temporal neighbor sampling (Fey et al., 2024).

### 3.3. Full-Resolution Property and Limitations of GSL

To better characterize the full-resolution property and facilitate a more principled analysis, we formalize it from an information-theoretic perspective as follows:

**Definition 3.2** (Full Resolution from an Information-Theoretic Perspective). Let $\mathcal{X}$ denote a RDB consisting of tables and FPKs. A REG $\mathcal{G}$ constructed from the schema graph is defined as *full-resolution* with respect to $\mathcal{X}$ if and only if:

$$H(\mathcal{X} \mid \mathcal{G}) = 0. \tag{1}$$

This condition implies that $\mathcal{G}$ preserves all information contained in $\mathcal{X}$ through the schema graph, ensuring that the RDB can be uniquely recovered from $\mathcal{G}$ without uncertainty.

To satisfy full-resolution, schema graph learning in RDL can differ from traditional GSL. Since the graph serves

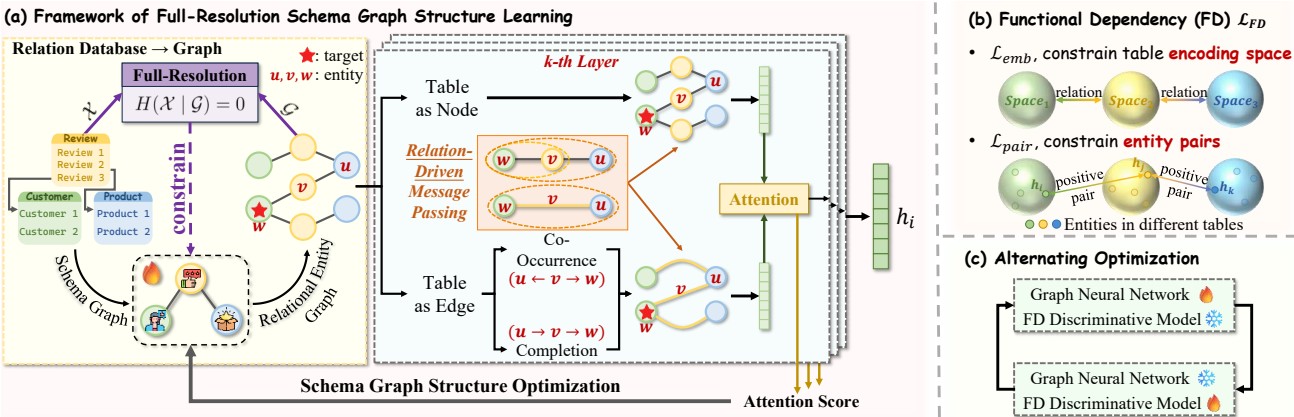

*Figure 4.* Overview of the proposed FROG method. (a) Through table-as-node and table-as-edge modeling with relation-aware message passing, the roles of tables under specific relational paradigms in the Schema Graph are learned, enabling REG structural optimization under full-resolution settings. (b) FD constraints, which regularize table-level embedding spaces and entity-level representations via discriminative model. (c) Alternating optimization between (a) and (b).

as a modeling proxy of the RDB rather than the primary data object, any structural modification should preserve the information required to recover the original relational state. We prove that two operations including edge pruning and edge augmentation in GSL violate the full-resolution property (see Appendix B.2).

**Proposition 3.3** (Non-Invertibility of Edge Pruning). *Let $\mathcal{G}_{orig} = (V, E)$ be the initial full-resolution graph of the RDB $\mathcal{X}$. Let $\Phi_{prune} : \mathcal{G}_{orig} \rightarrow \mathcal{G}_{sub}$ be an edge pruning operator such that $\mathcal{G}_{sub} = (V, E')$ with $E' \subset E$. If the pruning criterion is not preserved as part of the graph's metadata, then $\mathcal{G}_{sub} = \Phi_{prune}(\mathcal{G}_{orig})$ is not full-resolution:*

$$H(\mathcal{X} \mid \mathcal{G}_{sub}) > 0. \tag{2}$$

**Proposition 3.4** (Indistinguishability of Edge Addition). *Define $\mathcal{G}_{orig}$ as in Proposition 3.3. Let $\Phi_{add} : \mathcal{G}_{orig} \rightarrow \mathcal{G}_{aug}$ be a deterministic edge addition operator that generates an augmented graph $\mathcal{G}_{aug} = (V, E \cup \Delta E)$, where $\Delta E$ represents inferred edges. If $E$ and $\Delta E$ are indistinguishable in the output graph, then $\mathcal{G}_{aug} = \Phi_{add}(\mathcal{G}_{orig})$ is not full-resolution:*

$$H(\mathcal{X} \mid \mathcal{G}_{aug}) > 0. \tag{3}$$

### 3.4. Co-Optimization of REG and GNN Models

As discussed in Section 1, a predefined REG may hinder the model from effectively extracting features that are critical for downstream tasks, potentially limiting performance. To address this issue, jointly optimizing the graph structure and model parameters can be formulated as:

$$\min_{\theta \in \Theta} \mathcal{L}(\mathcal{M}(G)) \quad \text{s.t. } G \in \mathcal{G}_{\text{RDB}}^{\text{full}}. \tag{4}$$

Here, $\mathcal{G}_{\text{RDB}}^{\text{full}}$ denotes the set of graphs that satisfy the *full-resolution* property with respect to the RDB. To characterize

this property more concretely, we require that $G \in \mathcal{G}_{\text{RDB}}^{\text{full}}$ be capable of fully reconstructing the original RDB structure.

## 4. Method

In this section, We propose **FROG**, a *Full-Resolution and Optimizable Graph Structure Learning* framework for RDL. We first show that optimizing table roles preserves the full-resolution property and relate it to graph structure learning problem (▷ Sec. 4.1). Building on this, we analyze the modeling differences between table-as-node and table-as-edge instantiations and propose a framework for embedding-level schema graph optimization (▷ Sec. 4.2). In particular, we design distinct information propagation methods for different relational paradigms to better capture relational semantics in RDBs (▷ Sec. 4.3). Finally, we introduce the FD loss constraints at both table-level and the entity-level, encouraging the learned representations to respect the inherent dependencies of RDBs (▷ Sec. 4.4). Implementation details can be found in the Appendix E.

### 4.1. Table-as-Node & Table-as-Edge: A Full-Resolution Graph Structure Learning Method

When modeling an RDB as a graph, the full-resolution property is typically considered to preserve relational semantics (Fey et al., 2024). However, as stated in Sec. 3.3, GSL methods that directly manipulate graph edges are not suitable for this scenario. Inspired by Choi et al. (2025), table-as-edge in the schema graph offers an alternative for modeling relational semantics. In this setting, we prove that table-as-node and table-as-edge satisfy the full-resolution property under preserved entity and FPK types, features, and directions (see Appendix B.3). Fig. 5 illustrates how table-as-edge preserves the full-resolution property.

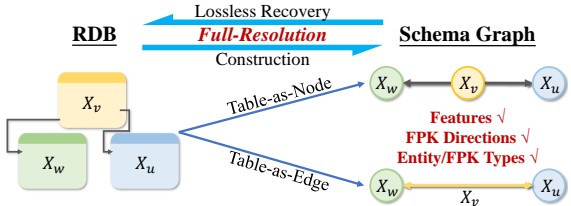

*Figure 5.* Illustration of table-as-node and table-as-edge

Building on this result, we reinterpret the GSL objective *from controlling edge existence to determining the roles of original tables*, both of which govern information propagation during message passing. This enables graph structure learning from an alternative perspective. Theorem 4.1 establishes that FROG based on table-as-node and table-as-edge continues to satisfy the full-resolution property, serving as a structural inductive bias to preserve relational semantics:

**Theorem 4.1** (Full-Resolution of Adaptive Mapping). *Let $\mathcal{T} = \{T_1, T_2, \ldots, T_k\}$ be a set of relational tables and $\mathcal{R}$ be the set of all FPK relations. Let $g : T_i \to \{f_n, f_e\}$ be a self-learning decision function that assigns each table $T_i$ to either a Table-as-Node mapping ($f_n$) or a Table-as-Edge mapping ($f_e$). The combined mapping $\mathcal{F}_{adaptive}$ : $(\mathcal{T}, \mathcal{R}) \to (V, E, X_V, X_E)$ is full-resolution.*

Since tables do not exist in isolation but are connected via FPK relations, a table may play different roles in different relational contexts. Motivated by this observation, we further refine graph structure learning in RDL by determining the roles of tables *under specific relations*, enabling the extraction of finer-grained structural information.

### 4.2. Node or Edge? From Graph Construction to Message Passing Optimization

Despite the benefits of role-based table modeling, a fundamental challenge remains: "table-as-node" or "table-as-edge" is discrete and mutually exclusive, precluding continuous interpolation between entity and relation-like semantics.

Crucially, this distinction is not merely structural in schema graph, but operational in GNNs: modeling a table directly affects the accessible information in message passing process. To make this distinction explicit, consider a relational dependency where information flows from entity $u$ to a target entity $w$ via an intermediate entity $v$ ($w - v - u$). If $w$ needs to obtain information from $u$, then: ① **Table-as-Node:** Require two rounds of aggregation, *i.e.*, $h_w^{(2),N} = f_n^2(h_v^{(1),N}) = f_n^2(f_n^1(X_u))$. ② **Table-as-Edge:** In a single aggregation step, corresponding to $h_w^{(1),E} = f_e(X_u)$. Here, $h_w^{(k),N/E}$ denotes the representation of $w$ after the $k$-th convolution when the intermediate entity $u$ is modeled as a node (N) or as an edge (E).

The following theorem formally characterizes their differ-

ences from an information-theoretic perspective.

**Theorem 4.2** (Information Retention Advantage). *Let $\mathcal{C}$ be the global context, and $f_n^2 \circ f_n^1 \in \mathcal{F}_N, f_e \in \mathcal{F}_E$ be the aggregation functions in hypothesis spaces, respectively. Assume that the embedding dimension is sufficiently large to avoid capacity-induced bottlenecks. If the intermediate node aggregation in the table-as-node strategy is not information-preserving, then modeling relational information as an edge allows the target entity to retain strictly more information about the source entity:*

$$\sup_{f_e} I(h_w^{(1),E}; X_u \mid \mathcal{C}) > \sup_{f_n^2 \circ f_n^1} I(h_w^{(2),N}; X_u \mid \mathcal{C}). \quad (5)$$

The analysis of Theorem 4.2 are provided in Appendix C. This reveals that the advantage of table-as-edge modeling lies in *avoiding intermediate aggregation bottlenecks*.

Building on Theorem 4.2, we reinterpret the optimization of table-as-node/edge paradigm *from graph construction to embedding fusion across different message-passing paths*. Specifically, we introduce a role-aware gating model $\mathcal{M}^{att}$ that adaptively balances the information from table-as-node and table-as-edge during message passing:

$$\tilde{g}_{v,r}^{(l)} = \sigma\left(\mathcal{M}_r^{att}\left([h_w^{(l),N} || h_w^{(l),E}]\right)\right). \quad (6)$$

This gate reflects the relative importance of edge-level relational information from table-as-edge paths ($u - v - w$) compared to node-level aggregation ($v - w$) under relation $r$, with respect to updating the representation of $w$.

To stabilize training, we smooth the adaptive gates by Exponential Moving Average (EMA) (Holt, 2004), after which the values are aggregated to obtain table-level gates:

$$
\begin{aligned}
g_{v,r} &= (1 - \alpha)\tilde{g}_{v,r}^{(l)} + \alpha \bar{g}_r, \\
\bar{g}_r &\leftarrow \mathbb{E}\left[g_{v,r}\right].
\end{aligned}
\quad (7)
$$

Here, $\bar{g}_r$ captures the average gating behavior, reflecting the learned role of the intermediate table $T$ under relation $r$.

Finally, the updated representation of entity $w$ at layer $l$ is obtained by combining node-level and edge-level messages weighted by the relation-conditioned role gate:

$$h_i^{(l)} = \sum_r (1 - \bar{g}_r) \cdot h_i^{(l),N} + \bar{g}_r \cdot h_i^E. \quad (8)$$

### 4.3. Relation-Driven Message Passing for RDL

Having established the optimization framework of FROG, we now describe the feature learning mechanism. Specifically, we introduce the relation-driven graph convolution operator that defines how node representations are computed under table-as-node and table-as-edge formulations.

Under the table-as-node formulation in RDBs, relationships are modeled as directed connections between entities from different tables, following a single $u \to v$ pattern. In this case, considering only node and edge types is sufficient, and standard heterogeneous GNNs can be directly applied.

In contrast, table-as-edge formulation is inherently more complex, as it involves interactions among more than two tables rather than a single table pair. Accordingly, we identify two representative patterns:

$$
\begin{aligned}
\text{CO-OCCURRENCE} &: u \leftarrow v \to w \ \Rightarrow \ u \overset{v}{\longleftrightarrow} w, \\
\text{COMPLETION} &: u \to v \to w \ \Rightarrow \ u \overset{v}{\longrightarrow} w.
\end{aligned}
\tag{9}
$$

The reasons for not adopting other table-as-edge paradigm types can be found in Appendix D. We now elaborate on the design of the aggregation mechanisms within the two table-as-edge relation paradigms proposed in this work.

**Co-Occurrence Type:**   In the Co-Occurrence paradigm, $u$ and $w$ are observed together in $v$ (e.g., user $\leftarrow$ review $\to$ product), meaning they play *symmetric roles* in $v$, and their interaction is explicitly captured through $v$. In the context of $v$, we model the message from $u$ to $w$ to capture their joint co-occurrence:

$$
\begin{aligned}
m_{u \to w} &= f_e^{u \leftarrow v \to w}(h_w, h_v, h_u) \\
&= W\left(h_w \| h_v \| h_u\right),
\end{aligned}
\tag{10}
$$

where $\|$ denotes concatenation and $W$ is a learnable matrix.

**Completion Type:**   In the Completion paradigm, the message from $u$ to $w$ is mediated by the entity $v$, which acts as *an intermediate that modulates the influence* of $u$ on $w$ (e.g., standing $\to$ race $\to$ circuit in Formula One racing competition). Formally, the message is defined as:

$$
\begin{aligned}
m_{u \to w} &= f_e^{u \to v \to w}(h_w, h_v, h_u) \\
&= W_2\left(h_w \| \sigma(f(h_v \| h_u)) \cdot W_1(h_v \| h_u)\right),
\end{aligned}
\tag{11}
$$

where $\|$ denotes concatenation, $\sigma$ is a gating function (e.g., sigmoid), and $W_1, W_2$ are learnable projection matrices.

These two message-passing mechanisms endow the model with the ability to perceive relational patterns within table-as-edge. Furthermore, to enable the model to be sensitive to relationships connecting different tables, we assign a dedicated GNN network to each possible relation type.

### 4.4. Functional Dependency Constraints in RDBs

Functional dependency (FD) constraints tables are essential for preserving relational semantics in RDBs. For instance, a review uniquely determines its associated customer and product. To maintain this inherent semantic consistency, the *FD Constraints should be taken into account* during the graph construction and model learning process.

In our setting, since all tuples within a table are jointly encoded, we do not consider intra-table feature-level FDs. Instead, we focus on FDs between entities across tables.

**FD Constraints in Graph Structure.**   As analyzed in Section 4.1, both *table-as-Node* and *table-as-Edge* satisfy the full-resolution property for RDBs, *i.e.*, the constructed graph can be fully recovered back to the original RDB without information loss. Consequently, FD constraints are preserved at the graph-structural level.

**FD Constraints in Graph Representation.**   While FD constraints are structurally preserved, we further aim to encode such dependencies in the learned representations. To this end, we interpret FDs as relations between entity representations across tables, and adopt embedding differences as a differentiable proxy for relational transformations:

$$
\begin{aligned}
\text{relation}_{ij} &\overset{\texttt{proxy}}{\longleftarrow} \text{diff}_{ij} = h_j - h_i, \\
\text{node}_i &\in T_p, \ \text{node}_j \in T_q, \quad \langle T_p, T_q \rangle \in \mathcal{R}.
\end{aligned}
\tag{12}
$$

This yields a differentiable and compatible with end-to-end graph learning. We implement the constraints in two complementary levels: the table-level and the entity-level.

First, we focus on modeling relationships at the table-level by regularizing the embedding differences associated with each table-to-table relation. In RDBs, a FD implies a consistent transformation pattern between entities linked by the same table relation (e.g., review→Product). Instead of constraining absolute entity representations, we assume that embedding differences induced by the same relation lie in a *shared low-dimensional relational subspace*.

Accordingly, we introduce a learnable projection matrix $P$ and enforce FD-related difference vectors to be well estimated with the low rank subspace spanned by $PP^T$. In addition, to identify the inherent asymmetry of relations, we introducing a learnable vector $s$. This leads to the table-level embedding space regularization in Eq. (13):

$$
\begin{aligned}
\mathcal{L}_{emb} &= \left\| (\text{diff}_{ij} - s) - PP^T(\text{diff}_{ij} - s) \right\|^2 \\
P &\in \mathbb{R}^{n \times d}, d < n
\end{aligned}
\tag{13}
$$

While the table-level constraint captures relational patterns shared by a give table-to-table dependency, it does not determine how individual entity-pairs should be distinguished within this pattern (e.g., review$_1$ → Product$_1$? or review$_1$ → Product$_2$?). We therefore introduce an entity-level constraint that operates on entity pairs.

For a given table-to-table dependency, we introduce a relation specific scoring model $\mathcal{M}^S$ to discriminate whether an entity pair actually satisfies the corresponding dependency:

$$
\text{score}_{\langle i,j \rangle} = \mathcal{M}^S_{(T_p, T_q)}(\text{diff}_{ij}) = \mathcal{M}^S_{(T_p, T_q)}(h_i - h_j).
\tag{14}
$$

*Table 1.* Performance comparison on the **Entity Classification** task measured by ROC-AUC (↑). The best results are highlighted in **bold**, and the second-best results are underlined.

| Dataset | Task | LightGBM | RDL | RelGNN | FROG |
|---|---|---|---|---|---|
| rel-avito | user-visits | $52.82_{\pm0.25}$ | $66.35_{\pm0.17}$ | $65.67_{\pm0.12}$ | $\mathbf{66.42}_{\pm0.14}$ |
| | user-clicks | $54.65_{\pm0.59}$ | $64.87_{\pm0.42}$ | $66.31_{\pm0.66}$ | $\mathbf{67.22}_{\pm0.25}$ |
| rel-event | user-repeat | $67.83_{\pm1.70}$ | $75.90_{\pm2.00}$ | $78.15_{\pm0.10}$ | $\mathbf{79.11}_{\pm0.87}$ |
| | user-ignore | $78.88_{\pm0.93}$ | $81.92_{\pm0.48}$ | $82.55_{\pm0.45}$ | $\mathbf{85.49}_{\pm0.10}$ |
| rel-f1 | driver-dnf | $65.26_{\pm4.74}$ | $72.73_{\pm1.07}$ | $73.45_{\pm0.64}$ | $\mathbf{74.39}_{\pm0.85}$ |
| | driver-top3 | $69.52_{\pm4.08}$ | $78.31_{\pm1.20}$ | $82.21_{\pm1.42}$ | $\mathbf{82.73}_{\pm0.51}$ |
| rel-trial | study-outcome | $70.70_{\pm0.50}$ | $68.63_{\pm0.34}$ | $69.43_{\pm0.46}$ | $\mathbf{70.92}_{\pm0.21}$ |

*Table 2.* Performance comparison on the **Entity Recommendation** task measured by MAP (↑). The best results are highlighted in **bold**, and the second-best results are underlined.

| Dataset | Task | LightGBM | RDL (ID-GNN) | RelGNN | FROG |
|---|---|---|---|---|---|
| rel-avito | user-ad-visit | $0.05_{\pm0.00}$ | $3.73_{\pm0.02}$ | $3.14_{\pm0.61}$ | $\mathbf{3.87}_{\pm0.04}$ |
| rel-hm | user-item-purchase | $0.34_{\pm0.03}$ | $2.78_{\pm0.01}$ | $2.76_{\pm0.01}$ | $\mathbf{2.86}_{\pm0.01}$ |
| rel-stack | user-post-comment | $0.04_{\pm0.01}$ | $12.98_{\pm0.21}$ | $13.54_{\pm0.18}$ | $\mathbf{13.58}_{\pm0.19}$ |
| | post-post-related | $1.98_{\pm0.61}$ | $10.81_{\pm0.15}$ | $11.16_{\pm0.00}$ | $\mathbf{11.48}_{\pm0.11}$ |
| rel-trial | condition-sponsor-run | $4.53_{\pm0.10}$ | $10.24_{\pm0.06}$ | $11.05_{\pm0.10}$ | $\mathbf{11.20}_{\pm0.07}$ |
| | site-sponsor-run | $9.63_{\pm0.31}$ | $18.91_{\pm0.03}$ | $\mathbf{19.10}_{\pm0.09}$ | $19.09_{\pm0.03}$ |

*Table 3.* Performance comparison on the **Entity Regression** task measured by MAE (↓). The best results are highlighted in **bold**, and the second-best results are underlined.

| Dataset | Task | Entity Mean | Entity Median | LightGBM | RDL | RelGNN | FROG |
|---|---|---|---|---|---|---|---|
| rel-avito | ad-ctr | 0.0462 | 0.0462 | $0.0408_{\pm0.0002}$ | $0.0413_{\pm0.0011}$ | $0.0393_{\pm0.0008}$ | $\mathbf{0.0385}_{\pm0.0002}$ |
| rel-event | user-attendance | 0.3035 | 0.2692 | $0.2636_{\pm0.0002}$ | $0.2582_{\pm0.0058}$ | $0.2417_{\pm0.0009}$ | $\mathbf{0.2408}_{\pm0.0022}$ |
| rel-fl | driver-position | 8.5013 | 8.5185 | $4.0685_{\pm0.0387}$ | $4.2074_{\pm0.1851}$ | $3.9807_{\pm0.0841}$ | $\mathbf{3.8906}_{\pm0.0695}$ |
| rel-hm | item-sales | 0.1105 | 0.0780 | $0.0761_{\pm0.0000}$ | $\mathbf{0.0556}_{\pm0.0004}$ | $0.0564_{\pm0.0010}$ | $0.0561_{\pm0.0001}$ |
| rel-stack | post-votes | 0.1061 | 0.0694 | $0.0679_{\pm0.0000}$ | $\mathbf{0.0652}_{\pm0.0001}$ | $0.0656_{\pm0.0002}$ | $0.0655_{\pm0.0001}$ |
| rel-trial | study-adverse | 57.9303 | 57.9303 | $\mathbf{42.8452}_{\pm0.7798}$ | $46.1570_{\pm0.1783}$ | $46.6503_{\pm0.4006}$ | $44.7536_{\pm0.9047}$ |
| | site-success | 0.4480 | 0.4411 | $0.4237_{\pm0.0020}$ | $0.4163_{\pm0.0082}$ | $0.3597_{\pm0.0164}$ | $\mathbf{0.3504}_{\pm0.0222}$ |

To encourage instance-wise separability, we adopt a contrastive objective based on the InfoNCE loss (Oord et al., 2018), which assigns higher scores to FD-consistent ($\langle pos \rangle$) pairs than to inconsistent ($\langle neg \rangle$) ones:

$$\mathcal{L}_{pair} = -\log \frac{d(\text{score}_{\langle pos \rangle})}{d(\text{score}_{\langle pos \rangle}) + \sum_k d(\text{score}_{\langle neg \rangle_k})}, \quad (15)$$
$$d(x) = \exp(x/\tau).$$

This objective enforces FD constraints in a manner consistent with entity relationships in relational databases.

We combine the table- and entity-level objectives using weights $\beta$ and $\gamma$ to balance their contributions (illustrated in Fig. 6), resulting in the final FD regularization term:

$$\mathcal{L}_{FD} = \beta \mathcal{L}_{emb} + \gamma \mathcal{L}_{pair}. \quad (16)$$

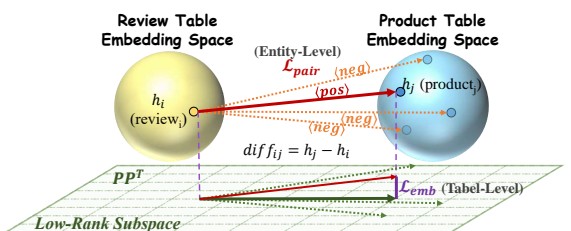

*Figure 6.* Illustration of $\mathcal{L}_{FD}$

The FD constraints are optimized alternating manner with GNN learning, ensuring that relational dependencies are enforced throughout the representation learning process. During GNN training, the low-rank matrix $P$ and the scoring models are kept fixed, and the overall loss is defined as:

$$\mathcal{L}_{main} = \mathcal{L}_{task} + \mathcal{L}_{FD} \quad (17)$$

## 5. Experiment

We design experiments to answer the following questions: **RQ1:** Does our method outperform the current GNN baselines? (▷ Sec. 5.2) **RQ2:** How does Table-as-Node/Edge affect model performance? (▷ Sec. 5.3) **RQ3:** How does FD loss restrict the embedding space between different tables? (▷ Sec. 5.4) **RQ4:** What information can we gain from the learned graph structure? (▷ Sec. 5.5) **RQ5:** Can the structure learned by the model on the current dataset be transferred to other tasks? (▷ Sec. 5.6)

### 5.1. Experimental Setup

**Datasets and Tasks.** We evaluate our FROG on Relbench (Robinson et al., 2024), a benchmark suite consisting of large-scale, real-world relational databases for machine learning research. Relbench covers diverse application domains and supports multiple predictive settings on relational data, including entity classification, regression, and link prediction (recommendation). Following the official benchmark protocol, we adopt ROC-AUC (Hanley & McNeil, 1983) for classification tasks, Mean Absolute Error

(MAE) for regression tasks, and Mean Average Precision (MAP)@K for recommendation tasks as evaluation metrics.

To comprehensively evaluate the effectiveness and generalization ability of our method, we conduct experiments on 6 representative datasets spanning 23 downstream tasks. Detailed information can be found in Appendix A.1.

**Baselines.** For baselines, we consider three categories of methods for relational deep learning: ① Statistical learning methods, including Entity Mean and Entity Median. ② Machine learning methods, specifically LightGBM (Ke et al., 2017), a widely used gradient boosting framework for heterogeneous tabular data. ③ GNN-based RDL methods, including the GraphSAGE (Hamilton et al., 2017) and ID-GNN (You et al., 2021) (used only for link prediction) implementation in Relbench (Robinson et al., 2024), as well as RelGNN (Chen et al., 2025), a representative GNN framework specifically designed for relational databases.

Detailed information about baselines and implement methods can be found in Appendix A.2.

### 5.2. Results Across Datasets and Tasks

We comprehensively evaluated the capabilities of FROG on 23 different tasks across 6 datasets. For each task, we run five trials and report the mean and standard deviation of the results. Complete results are reported in Appendix F.2.

**Results:** As shown in Table 1, 2 and 3, FROG achieves strong performance across different tasks and datasets. These results indicate that the role assigned to tables plays an important part, and that jointly optimizing graph structures and GNN models enables more expressive modeling, providing a competitive and effective framework for RDL.

### 5.3. Ablation Study: Node- and Edge-based Structure Modeling

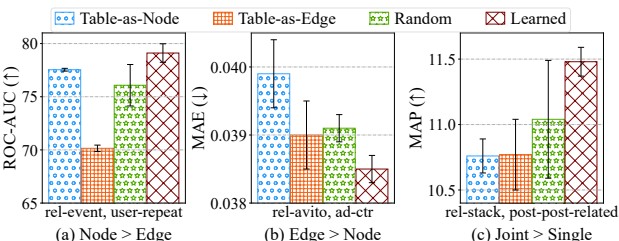

*Figure 7.* Experimental results about Table-as-Node or -Edge

To further investigate the impact of modeling tables-as-node or tables-as-edge under specific relations, we design three model variants to analyze the effects of different table roles: ① Modeling all tables as nodes; ② Modeling all eligible tables as edges whenever possible; and ③ Using a random weight to balance the table-as-node and table-as-edge repre-

sentations. Detailed results are provided in Appendix F.3.

**Results:** The results in Fig. 7 indicate that the role assigned to tables has a substantial impact on model performance. Depending on the specific setting, modeling tables as nodes, as edges, or jointly can be preferable. Nevertheless, our structure learning approach consistently achieves the best performance, highlighting the effectiveness of learning graph structures under the full-resolution restriction.

### 5.4. FD Constraints in Embedding Space

By introducing the $\mathcal{L}_{FD}$ loss, FROG enforces FD constraints in the embedding space. To examine this effect, we compute and visualize $\text{diff}_{ij}$ for entity pairs connected by FPK relations in RDBs with complex relations. We further focus on a single relation type and separately visualize pairs that satisfy the FD constraint and those that do not (*i.e.*, valid FPK pairs versus mismatched entity pairs). We select two datasets, rel-event, which contains multi-relations between the same pair of tables, and rel-avito, which exhibits more complex relational structures. The Schema Graphs of both datasets are provided in Appendix F.6.

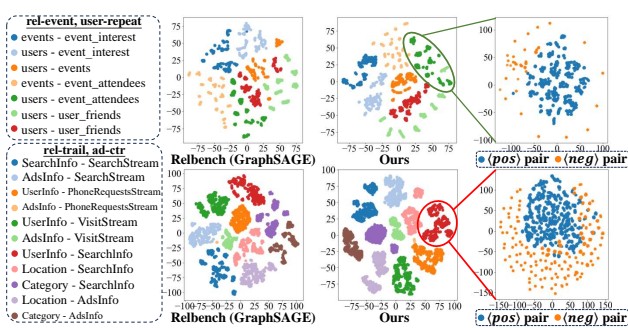

*Figure 8.* Visualization of relational patterns learned by the model

**Results:** The results in Fig. 8 show that our method yields more compact and better-separated relational representations, supporting the effectiveness of $\mathcal{L}_{emb}$. Moreover, for each relation type, the visualization reveals that matched entity pairs form tightly clustered distributions, while mismatched pairs are more dispersed, highlighting the role of $\mathcal{L}_{pair}$. Together, these observations indicate that the proposed losses jointly enforce functional dependency constraints in the learned representations. For more detailed numerical results, the ablation study on the $\mathcal{L}_{FD}$ can be found in Appendix F.5

### 5.5. Statistical Properties of the Learned Structure

After training, in addition to the model parameters, our approach also yields the learned graph structure. In this section, we conduct a statistical analysis to investigate the role patterns of tables across different relation types. Detailed results across datasets are provided in the Appendix F.7.

**Results:** From the empirical results shown in Fig. 9, we make the following key observations:

① In the Co-Occurrence paradigm, intermediate nodes tend to be modeled as *nodes* for classification and regression tasks, while they are more naturally modeled as *edges* for link prediction tasks.

② In the Co-Occurrence paradigm, intermediate node's roles are more likely to *remain consistent* in different directions (*i.e.* , $(u \to w)$ & $(w \to u)$), thereby preserving stable semantic relationships during message passing.

③ In the Completion paradigm, intermediate tables exhibit a clearer and more consistent preference for modeling as node or edge roles across different datasets.

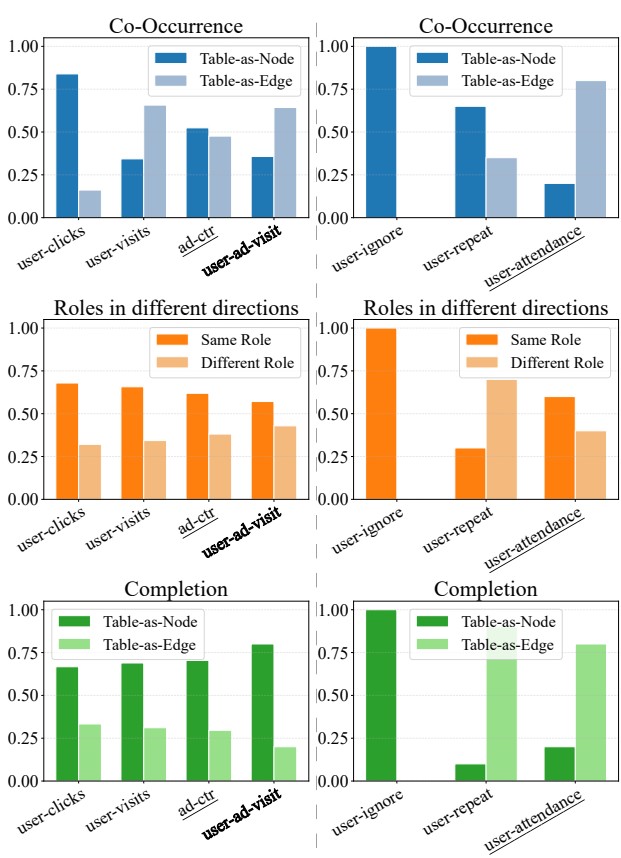

*Figure 9.* Structure Comparison Across Different Task Types on rel-avito (Left) and rel-event (Right), Including Classification, Regression, and **Link Prediction** Tasks

### 5.6. Structural Transferability

To study the transferability of learned structures across tasks, we conduct experiments on each dataset by training each task with structures learned from other tasks. We report results on rel-event and rel-avito as shown in Tables 4 and 5, where the $(i, j)$-th entry denotes the performance of task $i$ using the structure learned from task $j$.

*Table 4.* Performance Comparison on rel-event with Different Learned Structures. The best results are highlighted in **bold**, and the second-best results are underlined. ▨ and ▨ denote metrics evaluated by ROC-AUC ↑ and MAE ↓.

| Task | user-repeat | user-ignore | user-attendance |
|---|---|---|---|
| user-repeat | **79.01** | 76.30 | 74.31 |
| user-ignore | 83.32 | **84.59** | 84.39 |
| user-attendance | 0.2635 | 0.2635 | **0.2435** |

*Table 5.* Performance Comparison on rel-avito with Different Learned Structures. The best results are highlighted in **bold**, and the second-best results are underlined. ▨, ▨ and ▨ denote metrics evaluated by ROC-AUC ↑, MAE ↓ and MAP ↑.

| Task | user-visits | user-clicks | ad-ctr | user-ad-visit |
|---|---|---|---|---|
| user-visits | 66.37 | 66.18 | 66.36 | **66.54** |
| user-clicks | 65.58 | **67.15** | 65.52 | 66.03 |
| ad-ctr | 0.0390 | **0.0380** | 0.0385 | 0.0386 |
| user-ad-visit | 3.84 | **3.88** | 3.86 | 3.87 |

**Results:** The results show that learned table roles exhibit effective cross-task transferability, achieving competitive performance even when transferred across tasks. In some cases, transferred structures even lead to improved performance, suggesting that structures learned from different tasks may capture complementary relational patterns and shared structural characteristics.

## 6. Conclusion

In this work, we propose **FROG**, a *Full-Resolution and Optimizable Graph Structure Learning* framework for RDL. We formalize the full-resolution requirement for graph modeling in RDL from an information-theoretic perspective, and show how it can be satisfied through table-as-node and table-as-edge representations. To make optimization tractable, we reinterpret structural choices as differences in message passing behavior within GNNs and design relation-driven convolution operators accordingly. Furthermore, we introduce the $\mathcal{L}_{FD}$ to enforce functional dependencies in RDBs. Extensive experiments demonstrate that FROG achieves competitive performance, offering a new perspective on jointly optimizing graph structures and GNNs for RDL.

## Acknowledgements

The corresponding author is Qingyun Sun. This work is supported by NSFC under grants No.62225202 and No.62302023, and by the Fundamental Research Funds for the Central Universities. We extend our sincere thanks to all reviewers for their valuable efforts.

## Impact Statement

This paper presents work whose goal is to advance the field of Machine Learning. There are many potential societal consequences of our work, none which we feel must be specifically highlighted here.

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

## Appendix Contents

# A. Datasets, Tasks and Baselines

## A.1. Datesets & Tasks

*Table 6.* Task Details

| Dataset | Task name | Task type | #Rows of training table | | | #Unique Entities | %train/test Entity Overlap |
|---------|-----------|-----------|-------|------------|------|------------------|----------------------------|
| | | | Train | Validation | Test | | |
| *rel-avito* | user-clicks | classification | 59,454 | 21,183 | 47,996 | 66,449 | 45.3 |
| | user-visits | classification | 86,619 | 29,979 | 36,129 | 63,405 | 64.6 |
| | ad-ctr | regression | 5,100 | 1,766 | 1,816 | 4,997 | 59.8 |
| | user-ad-visit | recommendation | 86,616 | 29,979 | 36,129 | 63,402 | 64.6 |
| *rel-event* | user-repeat | classification | 3,842 | 268 | 246 | 1,514 | 11.5 |
| | user-ignore | classification | 19,239 | 4,185 | 4,010 | 9,799 | 21.1 |
| | user-attendance | regression | 19,261 | 2,014 | 2,006 | 9,694 | 14.6 |
| *rel-f1* | driver-dnf | classification | 11,411 | 566 | 702 | 821 | 50.0 |
| | driver-top3 | classification | 1,353 | 588 | 726 | 134 | 50.0 |
| | driver-position | regression | 7,453 | 499 | 760 | 826 | 44.6 |
| *rel-hm* | user-churn | classification | 3,871,410 | 76,556 | 74,575 | 1,002,984 | 89.7 |
| | item-sales | regression | 5,488,184 | 105,542 | 105,542 | 105,542 | 100.0 |
| | user-item-purchase | recommendation | 3,878,451 | 74,575 | 67,144 | 1,004,046 | 89.2 |
| *rel-stack* | user-engagement | classification | 1,360,850 | 85,838 | 88,137 | 88,137 | 97.4 |
| | user-badge | classification | 3,386,276 | 247,398 | 255,360 | 255,360 | 96.9 |
| | post-votes | regression | 2,453,921 | 156,216 | 160,903 | 160,903 | 97.1 |
| | user-post-comment | recommendation | 21,239 | 825 | 758 | 11,453 | 59.9 |
| | post-post-related | recommendation | 5,855 | 226 | 258 | 5,924 | 8.5 |
| *rel-trial* | study-outcome | classification | 11,994 | 960 | 825 | 13,779 | 0.0 |
| | study-adverse | regression | 43,335 | 3,596 | 3,098 | 50,029 | 0.0 |
| | site-success | regression | 151,407 | 19,740 | 22,617 | 129,542 | 42.0 |
| | condition-sponsor-run | recommendation | 36,934 | 2,081 | 2,057 | 3,956 | 98.4 |
| | site-sponsor-run | recommendation | 669,310 | 37,003 | 27,428 | 445,513 | 48.3 |

Relbench (Robinson et al., 2024) offered a collection of large-scale, real-world datasets for machine learning on relational databases. Notably, it defines different tasks across datasets, including Entity Classification, Entity Regression, and Entity Recommendation. In this study, we select 6 datasets covering 23 tasks for evaluation. A detailed description of the datasets and tasks is provided below.

- *rel-avito*: Originating from the Avito Kaggle challenge, this dataset is collected from a large online advertisement platform and is designed to predict ad click behavior based on contextual information. It includes user search logs, ad attributes, and related data across categories such as real estate and vehicles.

    - user-visits: Predict whether a user is likely to browse more than one distinct Ads within the upcoming 4-day window.
    - user-clicks: Predict whether a user will generate click interactions on more than one Ads over the next 4 days.
    - ad-ctr: Predict the click-through rate of an Ad, conditioned on it being clicked within the next 4 days.
    - user-ad-visit: Predict the list of Ads a user will visit within the next 4 days.

- *rel-event*: Sourced from the Hangtime mobile app, this dataset contains user activity data, event metadata, demographic information, and social relationships, capturing how social connections relate to event participation. All data is fully anonymized, with no personally identifiable information included.

    - user-repeat: Predict whether a user will express intent to attend an event (responding "yes" or "maybe") in the next week, conditioned on their participation in at least one event during the preceding 14 days.
    - user-ignore: Predict whether a user will ignore more than two event invitations over the next 7 days.

- `user-attendance`: Predict the aggregate count of events for which a user will indicate attendance intent (responding "yes" or "maybe") within the upcoming 7-day window.

- *rel-f1*: This dataset provides comprehensive Formula 1 racing data and statistics since 1950, covering drivers, constructors, and circuits. It includes detailed circuit information and full historical season data, such as race results, standings, qualifying sessions, and pit stop records.

  - `driver-dnf`: Predict the likelihood of a driver failing to complete a race within the next month.
  - `driver-top3`: Predict whether a driver will achieve the top-3 in any race during the next month.
  - `driver-position`: Predict the mean finishing rank of a driver across all races scheduled in the next 2 months.

- *rel-hm*: Derived from H&M transaction logs, this dataset records large-scale purchasing behaviors within the global fashion retail sector. It details the bipartite interactions between customers and articles enriched with extensive side information such as product design attributes and user demographics to support preference modeling tasks.

  - `user-churn`: Predict whether a customer will enter a churn state (defined as zero transactional activity) in the subsequent week.
  - `item-sales`: Predict the e total sales for a specific article over the next week.
  - `user-item-purchase`: Predict the sequence of articles a customer is expected to purchase within the next 7 days.

- *rel-stack*: Extracted from the stats-exchange site, this dataset represents a reputation-based question-and-answer network. It includes detailed records of user activity such as posts and comments with raw text, edit histories, voting behavior, and related posts.

  - `user-engagement`: Predict if a user will demonstrate active engagement (via voting, posting, or commenting) at any point in the next 3 months.
  - `user-badge`: Predict whether a user will acquire a new badge within the next 3 months.
  - `post-votes`: Predict the number of votes each user post will receive in the next 3 months.
  - `user-post-comment`: Predict the specific existing posts on which a user will leave a comment in the next 2 years.
  - `post-post-related`: Predict the set of existing posts that will form a linkage to a target post within the next 2 years.

- *rel-trial*: Curated from the AACT initiative, this dataset consolidates protocols and results from global clinical studies. It links sponsors, medical conditions, and facilities with detailed trial attributes such as study designs and outcomes to facilitate predictive research in healthcare operations and risk assessment.

  - `study-outcome`: Predict whether a clinical trial will successfully meet its primary endpoint objectives within the next year.
  - `study-adverse`: Predict the incidence count of patients suffering from severe adverse events or mortality in a trial within the next year.
  - `site-success`: Predict the overall success rate associated with a specific trial site in the next year.
  - `condition-sponsor-run`: Predict which sponsors will be formally associated with a specific medical condition.
  - `site-sponsor-run`: Predict whether a sponsor will use a facility as a clinical trial site.

### A.2. Baselines

In our work, we select three categories of methods as our baselines:

(i) Statistical learning method.

- Entity Regression
  - Entity Mean: Compute the average label for each entity in the training set and uses it as the predicted value for that entity.
  - Entity Median: Compute the median label for each entity in the training set and uses it as the predicted value for that entity.

- Entity Recommendation
  - Global Popularity: Compute the top-$K$ most frequent target entities across the entire training table, and predict these globally popular targets for all source entities.
  - Past Visit: Compute a source-specific top-$K$ list of historically visited target entities from the training table and use them for prediction.

(ii) Machine learning method.

- LightGBM (Ke et al., 2017): A gradient-boosted decision tree model that efficiently handles large-scale tabular data. It supports categorical features natively, offers fast training through histogram-based optimization, and automatically handles feature interactions.

(iii) GNN-based RDL methods.

- RDL (GraphSAGE) (Hamilton et al., 2017; Robinson et al., 2024): A GraphSAGE-based heterogeneous graph neural network (Bing et al., 2023). For each relation type, a corresponding GNN is selected to perform message passing and convolution. The resulting representations are then aggregated to obtain node embeddings, which are fed into different downstream task-specific heads to produce the final predictions.
- RDL (ID-GNN) (You et al., 2021; Robinson et al., 2024): Only used in link prediction (recommendation) tasks. An identity-aware GNN (ID-GNN) is used as the backbone to capture node identity information. For each source entity, target entity embeddings computed by the GNN are fed into an MLP prediction head to produce pairwise scores. The ID-GNN is trained using the standard binary cross-entropy loss.
- RelGNN (Chen et al., 2025): RelGNN introduced the concept of atomic paths, which represent composite relations spanning multiple tables and encode richer semantic information. It performs convolution along atomic paths with multi-head attention (Vaswani et al., 2017), enabling the model to capture semantic information more comprehensively.

### A.3. Baseline Implementation Details

For GraphSAGE and ID-GNN, we trained them according to the hyperparameters specified in Relbench.

For RelGNN, we searched for the optimal learning rate while keeping all other hyperparameters consistent with those in the original paper, and reported the results under the optimal learning rate.

## B. Theoretical Analysis of Full-Resolution

### B.1. Full-Resolution from an Information Perspective

**Definition B.1** (Full Resolution from an Information-Theoretic Perspective). Let $\mathcal{X}$ denote a RDB consisting of tables and FPKs. A REG $\mathcal{G}$ constructed from the schema graph is defined as *full-resolution* with respect to $\mathcal{X}$ if and only if:

$$H(\mathcal{X} \mid \mathcal{G}) = 0. \tag{18}$$

This condition implies that $\mathcal{G}$ preserves all information contained in $\mathcal{X}$ through the schema graph, ensuring that the RDB can be uniquely recovered from $\mathcal{G}$ without uncertainty.

### B.2. Graph Structure Learning Without Structural Provenance Is NOT Full-Resolution

**Proposition B.2** (Non-Invertibility of Edge Pruning). *Let $\mathcal{G}_{orig} = (V, E)$ be the initial full-resolution graph of the RDB $\mathcal{X}$. Let $\Phi_{prune} : \mathcal{G}_{orig} \to \mathcal{G}_{sub}$ be an edge pruning operator such that $\mathcal{G}_{sub} = (V, E')$ with $E' \subset E$. If the pruning criterion is not preserved as part of the graph's metadata, then $\mathcal{G}_{sub} = \Phi_{prune}(\mathcal{G}_{orig})$ is not full-resolution:*

$$H(\mathcal{X} \mid \mathcal{G}_{sub}) > 0. \tag{19}$$

*Proof.* Since $\mathcal{G}_{orig}$ satisfies full-resolution with respect to $\mathcal{X}$, $H(\mathcal{X} \mid \mathcal{G}_{orig}) = 0$ holds. Moreover, as $\mathcal{G}_{orig}$ is constructed via $\mathcal{X}$, $H(\mathcal{G}_{orig} \mid \mathcal{X}) = 0$ follows. Hence, to prove $H(\mathcal{X} \mid \mathcal{G}_{sub}) > 0$, it suffices to show $H(\mathcal{G}_{orig} \mid \mathcal{G}_{sub}) > 0$.

Consider the pruning operator $\Phi_{prune} : (V, E) \to (V, E')$ with $E' \subset E$, where the pruning criterion is not retained as metadata. Then $\Phi_{prune}$ is not injective.

Specifically, let $E_0 \subset E$ and choose two distinct edges $e_i \neq e_j$ with $e_i, e_j \notin E_0$. Define:

$$\mathcal{G}_1 = (V, E_0 \cup \{e_i\}), \quad \mathcal{G}_2 = (V, E_0 \cup \{e_j\}). \tag{20}$$

Under the same pruning rule, both edges are removed, *i.e.* ,

$$\Phi_{prune}(\mathcal{G}_1) = \Phi_{prune}(\mathcal{G}_2) = (V, E_0) = \mathcal{G}_{sub}, \quad \mathcal{G}_1 \neq \mathcal{G}_2. \tag{21}$$

Since $|\Phi_{prune}^{-1}(\mathcal{G}_{sub})| \geq 2$, the original graph $\mathcal{G}_{orig}$ is not uniquely determined by $\mathcal{G}_{sub}$. Therefore, $\max_g \mathbb{P}(\mathcal{G}_{orig} = g \mid \mathcal{G}_{sub}) < 1$, which implies:

$$H(\mathcal{G}_{orig} \mid \mathcal{G}_{sub}) > 0. \tag{22}$$

$\square$

**Proposition B.3** (Indistinguishability of Edge Addition). *Let $\mathcal{G}_{orig} = (V, E)$ be the initial full-resolution graph of the RDB $\mathcal{X}$. Let $\Phi_{add} : \mathcal{G}_{orig} \to \mathcal{G}_{aug}$ be a deterministic edge addition operator that generates an augmented graph $\mathcal{G}_{aug} = (V, E \cup \Delta E)$, where $\Delta E$ represents inferred edges. If $E$ and $\Delta E$ are indistinguishable in the output graph, then $\mathcal{G}_{aug} = \Phi_{add}(\mathcal{G}_{orig})$ is not full-resolution:*

$$H(\mathcal{X} \mid \mathcal{G}_{aug}) > 0. \tag{23}$$

*Proof.* Sa in Proposition B.2, it is sufficient to show $H(\mathcal{G}_{orig} \mid \mathcal{G}_{aug}) > 0$. Since there are multiple different original graphs, the same extended graph can still be generated after adding edges. If original and inferred edges are indistinguishable, $\mathcal{G}_{aug}$ cannot uniquely determine $\mathcal{G}_{orig}$, which implies: $H(\mathcal{G}_{orig} \mid \mathcal{G}_{aug}) > 0$. $\square$

Together, these propositions show that structural learning which *only modifies graph topology without preserving the distinction between original and modified structures* is inherently information-losing and therefore cannot get a full-resolution representation.

### B.3. Full-Resolution of Table-as-Node and Table-as-Edge Operations

**Lemma B.4** (Table-as-Node Mapping is Full-Resolution). *Let $T = \{\tau_1, \tau_2, \ldots, \tau_n\}$ be a relational table where each tuple $\tau_i$ is associated with an attribute vector $A(\tau_i)$. Let $\mathcal{R}_T$ denote the set of Foreign-Primary Key (FPK) links involving tuples in $T$.*

*Define the table-as-node mapping $f_n : (T, \mathcal{R}_T) \to (V, E, X_V)$ such that:*

- *Each tuple $\tau_i \in T$ is mapped to a unique node $v_i \in V$ with features $x_{v_i} = A(\tau_i) \in X_V$, preserving the original node type.*
- *Each FPK relation $\ell_{i \to j} \in \mathcal{R}_T$ is mapped to a directed, typed edge $e_{i \to j} = (v_i, v_j) \in E$.*

*Then the mapping $f_n$ is full-resolution.*

*Proof.* Let's construct an explicit inverse mapping $\Psi_n$. Since $f_n$ maps each tuple $\tau_i$ to a unique node $v_i$ without loss of attribute information, the original tuple $\tau_i$ can be uniquely recovered from the node identity and its feature vector $x_{v_i}$. This establishes a bijection between $T$ and $V$.

Furthermore, each relation $\ell_{i \to j}$ is mapped to a directed edge $e_{i \to j}$ where the direction and type are preserved. Consequently, the complete set of relations $\mathcal{R}_T$ can be uniquely reconstructed from the edge set $E$ and the node mapping.

Since the inverse mapping $\Psi_n : (V, E, X_V) \to (T, \mathcal{R}_T)$ is deterministic and exists for any $(V, E, X_V)$ in the image of $f_n$, the conditional entropy satisfies:

$$H((T, \mathcal{R}_T) \mid f_n(T, \mathcal{R}_T)) = 0. \tag{24}$$

Thus, $f_n$ is full-resolution. $\square$

**Lemma B.5** (Table-as-Edge Mapping is Full-Resolution). *Let $T = \{\tau_1, \ldots, \tau_n\}$ be a relational table where each tuple $\tau_i$ has an attribute vector $A(\tau_i)$. Let $T_r = \{\tau'_1, \ldots, \tau'_m\}$ be a reference table (or a set of tables) connected to $T$ via Primary-Foreign Key (FPK) links $\mathcal{R}_T$.*

*Define the table-as-edge mapping $f_e : (T, \mathcal{R}_T, T_r) \to (V, E, X_V, X_E)$ as follows:*

- *Each tuple $\tau'_j \in T_r$ is mapped to a unique node $v_j \in V$ with features $x_{v_j} = A(\tau'_j)$.*

- *Each tuple $\tau_i \in T$ is mapped to an edge $e_i \in E$ with edge features $x_{e_i} = A(\tau_i)$.*

- *The incidence structure of each edge $e_i$ is determined by $\mathcal{R}_T$. For each FPK link between $\tau_i$ and a tuple $\tau'_j$, the mapping $f_e$ preserves the link's metadata, including the FPK directions, as properties of the incidence between $e_i$ and the corresponding node $v_j$.*

*Then the mapping $f_e$ is full-resolution.*

*Proof.* Let's construct an explicit inverse mapping $\Psi_e$ to demonstrate information preservation. Each tuple $\tau_i \in T$ is mapped to a unique edge $e_i \in E$ whose feature $A(\tau_i)$ is preserved as $x_{e_i}$.

Moreover, the incidence structure of $e_i$ uniquely specifies the set of tuples in $T_r$ referenced by $\tau_i$ via FPK $\mathcal{R}_T$. Since the direction of each FPK are preserved as part of metadata, the tuple $\tau_i$ can be uniquely recovered and all FPK relations in $\mathcal{R}_T$ can be reconstructed.

Similarly, each node $v_j \in V$ uniquely corresponds to a tuple $\tau'_j \in T_r$ via its node features.

Hence, a deterministic inverse mapping $\Psi_e : (V, E, X_V, X_E) \to (T, T_r, \mathcal{R}_T)$ exists, and

$$H((T, T_r, \mathcal{R}_T) \mid f_e(T, T_r, \mathcal{R}_T)) = 0. \tag{25}$$

This concludes that $f_e$ is full-resolution. □

## B.4. FROG is Full-Resolution

**Theorem B.6** (Full-Resolution of Adaptive Mapping). *Let $\mathcal{T} = \{T_1, T_2, \ldots, T_k\}$ be a set of relational tables and $\mathcal{R}$ be the set of all FPK relations. Let $g : T_i \to \{f_n, f_e\}$ be a self-learning decision function that assigns each table $T_i$ to either a Table-as-Node mapping ($f_n$) or a Table-as-Edge mapping ($f_e$). The combined mapping $\mathcal{F}_{adaptive} : (\mathcal{T}, \mathcal{R}) \to (V, E, X_V, X_E)$ is full-resolution.*

*Proof.* To prove that $\mathcal{F}_{adaptive}$ is full-resolution, we show that an inverse mapping $\Psi$ exists such that the original relational state can be uniquely recovered.

The decision function $g$ partitions the set of tables $\mathcal{T}$ into two disjoint subsets: $\mathcal{T}_{node} = \{T_i \mid g(T_i) = f_n\}$ and $\mathcal{T}_{edge} = \{T_j \mid g(T_j) = f_e\}$.

- For any $T_i \in \mathcal{T}_{node}$, by **Lemma B.4**, the mapping $f_n$ is full-resolution between tables and nodes, preserving all attributes and internal links.

- For any $T_j \in \mathcal{T}_{edge}$, by **Lemma B.5**, the mapping $f_e$ is full-resolution between tables and (hyper)edges, preserving all attributes and the incidence structure.

Observe that each table $T_k \in \mathcal{T}$ occupies a *disjoint representation domain*: tuples from different tables are never merged, aggregated, or identified under $\mathcal{F}_{adaptive}$. The adaptive mapping acts independently on each table according to the decision function $g$, and the resulting graph components are combined via set union.

As a consequence, the local inverse mappings $\Psi_n$ and $\Psi_e$ defined in Lemma B.4 and Lemma B.5 can be composed into a global deterministic inverse mapping:

$$\Psi = \left( \bigcup_{T_i \in \mathcal{T}_{node}} \Psi_n \right) \cup \left( \bigcup_{T_j \in \mathcal{T}_{edge}} \Psi_e \right), \tag{26}$$

which uniquely reconstructs the original RDB from the graph.

Therefore,

$$H((\mathcal{T}, \mathcal{R}) \mid \mathcal{F}_{adaptive}(\mathcal{T}, \mathcal{R})) = 0, \tag{27}$$

proving that the self-learning mapping is full-resolution. □

# C. Analysis and Proof of the Theorem 4.2

### C.1. Problem Setup

Consider a path $u - v - w$. Each node $x$ has raw features $X_n$, and its initial embedding is $h_n^{(0)} := X_n$.

We compare the amount of information about $X_u$ that becomes available at $w$ under two aggregation strategies, measured by conditional mutual information after removing the contextual information that can explain away such information. The two aggregation strategies are:

**Table-as-Node:**  Require two rounds of aggregation, *i.e.* $h_w^{(2),N} = f_n^2(h_v^{(1),N}) = f_n^2(f_n^1(X_u))$.

**Table-as-Edge:**  In a single aggregation step, node $v$ can directly access its two-hop neighbors' initial embeddings $h_w^{(1),E} = f_e(X_u)$.

Here, $h_w^{(k),N/E}$ denotes the representation of $w$ after the $k$-th convolution when the intermediate entity $u$ is modeled as a node (N) or as an edge (E).

### C.2. Global Context for Conditional MI

To isolate the influence of the surrounding graph context, we define the global context as:

$$\mathcal{C} = \{A, \{X_k \mid k \neq u\}\}, \tag{28}$$

where $A$ denotes the graph structure and $x_k$ denotes node features for all nodes except $u$. Under this definition, the only random variable outside the context is $X_u$.

### C.3. Conditional Markov Property

Before proving the theorem, a lemma is first introduced to clarify the conditional Markov chain, laying the groundwork for the proof.

**Lemma C.1** (Conditional Markov Property). *Under the conditioning on the global context $\mathcal{C}$, the following conditional Markov chain holds:*

$$X_u \ \rightarrow \ h_v^{(1),N} \ \rightarrow \ h_w^{(2),N} \qquad (conditioned \ on \ \mathcal{C}). \tag{29}$$

*Proof.* We prove the lemma by explicitly characterizing the dependency structure induced by two-round message passing under the conditioning on the global context $\mathcal{C}$.

Under the global context $\mathcal{C} = \{A, \{X_k \mid k \neq u\}\}$, the graph structure $A$ and all node features except $X_u$ are fixed. Consequently, the only remaining source of randomness in the computation is the node feature $X_u$.

In the first aggregation round, the representation at node $v$ is computed as:

$$h_v^{(1),N} = f_n^1(X_v, \ \{X_k : k \in \mathcal{N}(v)\}), \tag{30}$$

where $f_n^1$ is a deterministic aggregation function. Conditioned on $\mathcal{C}$, this expression reduces to a deterministic function of $X_u$ whenever $u \in \mathcal{N}(v)$.

In the second aggregation round, the representation at node $w$ is given by:

$$h_w^{(2),N} = f_n^2\Big(X_w, \ \{h_k^{(1),N} : k \in \mathcal{N}(w)\}\Big), \tag{31}$$

where $f_n^2$ is also a deterministic aggregation function. Under the conditioning on $\mathcal{C}$, all arguments of $f_n^2$ are fixed except those depending on $h_v^{(1),N}$.

Therefore, conditioned on $\mathcal{C}$, the second-round representation can be rewritten as:

$$h_w^{(2),N} = f_n\Big(h_v^{(1),N}, \mathcal{C}\Big), \tag{32}$$

for some deterministic function $f_n$ induced by $f_n^2$. Importantly, $X_u$ **does not appear as a direct argument of** $f_n$.

By the definition of conditional independence, this functional dependence implies

$$P\left(h_w^{(2),N} \mid h_v^{(1),N}, X_u, \mathcal{C}\right) = P\left(h_w^{(2),N} \mid h_v^{(1),N}, \mathcal{C}\right). \tag{33}$$

Hence, the conditional Markov chain:

$$X_u \;\rightarrow\; h_v^{(1),N} \;\rightarrow\; h_w^{(2),N} \qquad \text{(conditioned on } \mathcal{C}) \tag{34}$$

holds, which completes the proof. $\qquad\square$

### C.4. Information Retention Advantage

**Notice.** The core message of Theorem C.2 is **not** to compare the optimality of specific aggregation functions in Strategy A and Strategy B, but to **highlight a structural difference** in their information propagation mechanisms. Even under idealized function classes and sufficient representation capacity, Table-as-Node involves multiple sequential aggregation stages, **each of which introduces a potential source of irreversible information loss**. In contrast, Table-as-Edge shortens the information transmission path, thereby reducing the number of loss-inducing transformations.

**Theorem C.2** (Information Retention Advantage). *Let $\mathcal{C}$ be the global context, and $f_n^2 \circ f_n^1 \in \mathcal{F}_N, f_e \in \mathcal{F}_E$ be the aggregation functions in hypothesis spaces, respectively. Assume that the embedding dimension is sufficiently large to avoid capacity-induced bottlenecks. If the intermediate node aggregation in the table-as-node strategy is not information-preserving, then modeling relational information as an edge allows the target entity to retain strictly more information about the source entity:*

$$\sup_{f_e} I(h_w^{(1),E}; X_u \mid \mathcal{C}) > \sup_{f_n^2 \circ f_n^1} I(h_w^{(2),N}; X_u \mid \mathcal{C}). \tag{35}$$

*Proof.* Under Table-as-Node paradigm, for a fixed context $\mathcal{C}$, the computation of $h_w^{(2),N}$ follows the sequential mapping: $h_w^{(2),N} = f_n^2(h_v^{(1),N}, \mathcal{C})$, where $h_v^{(1),N} = f_n^1(X_u, \mathcal{C})$. This induces a *conditional Markov chain*:

$$X_u \rightarrow h_v^{(1),N} \rightarrow h_w^{(2),N} \qquad \text{(conditioned on } \mathcal{C}). \tag{36}$$

By the conditional version of the Data Processing Inequality (DPI), for any $f_n^1, f_n^2 \in \mathcal{F}_N$, we have:

$$I(h_w^{(2),N}; X_u \mid \mathcal{C}) \leq I(h_v^{(1),N}; X_u \mid \mathcal{C}). \tag{37}$$

Since the first-hop aggregation $f_n^1$ is not information-preserving, we have:

$$I(h_v^{(1),N}; X_u \mid \mathcal{C}) < H(X_u \mid \mathcal{C}). \tag{38}$$

Summarize them together:

$$\underbrace{I(h_w^{(2),N}; X_u \mid \mathcal{C}) \leq I(h_v^{(1),N}; X_u \mid \mathcal{C})}_{\text{DPI}} \overbrace{< H(X_u \mid \mathcal{C})}^{\text{Lossy Aggregation}}. \tag{39}$$

In Table-as-Edge paradigm, node $w$ directly accesses the 2-hop neighborhood $\mathcal{N}(w) \cup \mathcal{N}^2(v)$, which contains $u$. The aggregation is defined as $h_w^{(1),E} = f(X_u, \mathcal{C})$. By the property of MI, we have:

$$I(h_w^{(1),E}; X_u \mid \mathcal{C}) \leq H(X_u \mid \mathcal{C}). \tag{40}$$

If the embedding dimension is sufficiently large so that representation capacity is not the limiting factor for mutual information, then the hypothesis spaces $\mathcal{F}_N$ and $\mathcal{F}_E$ are rich enough to approximate arbitrary injective mappings. As a result, both strategies are able to attain their respective information-theoretic upper bounds.

Since the upper bound derived in Eq. (40) is strictly larger than that in Eq. (39) under the non-lossless first-hop aggregation assumption, it follows that:

$$\sup_{f_e} I(h_w^{(1),E}; X_u \mid \mathcal{C}) > \sup_{f_n^2 \circ f_n^1} I(h_w^{(2),N}; X_u \mid \mathcal{C}). \tag{41}$$

$\qquad\square$

## D. Choice of the Table-as-Node Paradigm

We do not explicitly include the pattern $u \to v \leftarrow w$ as one of the canonical propagation paradigms.

Before analyzing the underlying reasons, we first introduce two concepts: *dimensional tables and fact tables*.

Dimensional tables and fact tables play distinct roles in relational databases. Dimensional tables typically store descriptive attributes that provide contextual information about entities, such as user profiles or product metadata, and are often referenced by multiple other tables. In contrast, fact tables record transactional or event-level information, capturing interactions or measurements that link multiple dimensional tables, such as user-item interactions or event logs.

The reason is that the intermediate node $v$ in $u \to v \leftarrow w$ typically corresponds to a *dimensional table* referenced by multiple tables. Such dimensional tables typically encode static or descriptive attributes and act as shared reference points, rather than sources of relational signals that should be propagated across different fact tables. Propagating information through $v$ in this pattern effectively mixes representations of unrelated entities that merely share the same reference, which provides limited benefit for cross-entity representation learning. Therefore, in the table-as-edge formulation, we omit this pattern to simplify message aggregation and to avoid introducing spurious or redundant information propagation.

Nevertheless, our framework is scalable: once the information transmission paradigm is specified, it can be seamlessly incorporated into FROG.

## E. The Implement of FROG

### E.1. Pseudo Code for the Learning Process of the FROG Framework

In this section, we analyze the pseudocode of FROG to facilitate a clearer understanding of the proposed approach. Specifically, Alg. 1 illustrates how role representations of tables are learned under different relations, enabling embedding fusion and graph structure learning. Alg. 2 then provides a high-level description of the overall framework, with a particular emphasis on the alternating optimization strategy.

---

**Algorithm 1** GNN Aggregation Method of FROG

---

**Input:** Node feature $X = \{x_1, \cdots, x_n\}$, Table-as-Node Graph $G_N$, Table-as-Edge Graph $G_E$, Node Types $C$
**Output:** Node Embedding $H = \{h_1, \cdots, h_n\}$, Tables' Role in Specific Relations $\bar{g}_r$
Initialize $H^{(0)} = X, \bar{g}_r$.
**for** $l = 1$ **to** Model Layer **do**
  **for** $c$ **in** C **do**
    $H^{(l),N}[c] \leftarrow \text{GraphSAGE}_c(H^{(l-1)})$ with structure $G_N$    // e.g. $v - w$
    $H^{(l),E}[c] \leftarrow \text{GNN}_c(H^{(l-1)})$ via Eq.(10) and Eq.(11), with structure $G_E$    // e.g. $u \overset{v}{-} w$
  **end for**
  **for** $(\text{type}_u, \text{type}_v, \text{type}_w)$ **in** $\texttt{rel}_E$ **do**
    **for** $(\text{type}'_v, \text{type}'_w)$ **in** $\texttt{rel}_N$ **do**
      **if** $\text{type}_v = \text{type}'_v$ and $\text{type}_w = \text{type}'_w$ **then**
        Compute $\bar{g}_r$ via Eq.(6) and Eq.(7)
      **end if**
    **end for**
  **end for**
  **for** $c$ **in** C **do**
    Compute $H^{(l)}[c]$ via Eq.(8)
  **end for**
**end for**
**Return:** $H^{(l)}, \bar{g}_r$

---

---

**Algorithm 2** FROG Framework

---

**Input:** Node feature $X = \{x_1, \cdots, x_n\}$, Relations of Table-as-Node $\mathtt{rel}_N$ $(v - w)$, Relations of Table-as-Edge $\mathtt{rel}_E$
$(u - v - w)$, Node Types $C$, Epochs, GNN $\mathcal{M}$, FD discriminative mode $\mathcal{M}_{FD}$
Initialize $\mathcal{M}$, $\mathcal{M}_{FD}$.
Construct $G_N$ based $\mathtt{rel}_N$.      // Data preprocessing, consistent with other RDL methods.
Construct $G_E$ based $\mathtt{rel}_E$.     // $\mathcal{O}(|\mathtt{rel}_E||E| \log |E|)$
**for** $i = 1$ **to** Epochs **do**
  Fix $\mathcal{M}_{FD}$, Train $\mathcal{M}$
  Get Predictions of FROG via Alg.1
  Get $\mathcal{L}_{main}$ via Eq.(17).
  Optimize $\mathcal{M}$
  Fix $\mathcal{M}$, Train $\mathcal{M}_{FD}$
  Get $\mathcal{L}_{FD}$ via Eq.(16).
  Optimize $\mathcal{M}_{FD}$
**end for**

---

### E.2. Implement Details

We obtain initial node embeddings from raw table attributes using PyTorch Frame (Hu et al., 2024) and leverage GloVe (Pennington et al., 2014) to encode textual attributes, following the setting in Relbench (Robinson et al., 2024). We implement the GNN model based on PyTorch Geometric (Fey & Lenssen, 2019). To avoid test-time data leakage (Kapoor & Narayanan, 2023), we adopt temporal neighbor sampling during dataset splitting, using timestamp constraints to ensure proper training.

## F. Experiments

### F.1. Hyperparameter Selection

The experimental results were obtained via grid search, with Table 7 showing the parameter search ranges. In particular, during training, the number of samples for different-hop entities is controlled through the number of Neighbor Samples. Following the Relbench setup, the number of neighbors sampled at the $i$-th hop is set as Neighbor Samples$/2^i$.

*Table 7.* Hyper-parameter search space

| Parameter | Search space |
|---|---|
| Learning Rate | $\{0.005, 0.001, 0.0005, 0.0001\}$ |
| Neighbor Samples | $\{64, 128\}$ |
| Channels | $\{32, 64, 128\}$ |
| Layers | $\{1, 2, \ldots, 8\}$ |
| Dropout | $\{0.0, 0.2, 0.3, 0.5\}$ |

### F.2. Full Results of Main Experiments

As shown in Table 8, 9 and 10, the FROG outperforms or achieves performance comparable to the state of the art across different tasks on various datasets. Notably, for datasets that exhibit relatively poor performance on node-level tasks (classification and regression), FROG demonstrates strong performance on entity recommendation, such as rel-hm and rel-stack. This observation suggests that, by placing greater emphasis on relations, FROG is able to better model tasks involving interactions between node pairs.

### F.3. Ablation Study: Node or Edge

We investigate the impact of table roles in graph construction on the model's learning capability by evaluating three variants of FROG: ① modeling all tables as nodes; ② modeling all eligible tables as edges whenever possible; and ③ using a random weight to balance the table-as-node and table-as-edge representations.

*Table 8.* Performance comparison on the **Entity Classification** task measured by ROC-AUC (↑). The best results are highlighted in **bold**, and the second-best results are underlined.

| Dataset | Task | LightGBM | RDL | RelGNN | FROG |
|---------|------|----------|-----|--------|------|
| rel-avito | user-visits | $52.82_{\pm0.25}$ | $66.35_{\pm0.17}$ | $\underline{65.67}_{\pm0.12}$ | $\mathbf{66.42}_{\pm0.14}$ |
| | user-clicks | $54.65_{\pm0.59}$ | $64.87_{\pm0.42}$ | $66.31_{\pm0.66}$ | $\mathbf{67.22}_{\pm0.25}$ |
| rel-event | user-repeat | $67.83_{\pm1.70}$ | $75.90_{\pm2.00}$ | $\underline{78.15}_{\pm0.10}$ | $\mathbf{79.11}_{\pm0.87}$ |
| | user-ignore | $78.88_{\pm0.93}$ | $81.92_{\pm0.48}$ | $\underline{82.55}_{\pm0.45}$ | $\mathbf{85.49}_{\pm0.10}$ |
| rel-fl | driver-dnf | $65.26_{\pm4.74}$ | $72.73_{\pm1.07}$ | $\underline{73.45}_{\pm0.64}$ | $\mathbf{74.39}_{\pm0.85}$ |
| | driver-top3 | $69.52_{\pm4.08}$ | $78.31_{\pm1.20}$ | $\underline{82.21}_{\pm1.42}$ | $\mathbf{82.73}_{\pm0.51}$ |
| rel-hm | user-churn | $55.27_{\pm0.09}$ | $\underline{69.70}_{\pm0.06}$ | $\mathbf{70.64}_{\pm0.10}$ | $69.40_{\pm0.05}$ |
| rel-stack | user-engagement | $63.43_{\pm0.33}$ | $\underline{90.58}_{\pm0.10}$ | $\mathbf{90.69}_{\pm0.04}$ | $90.53_{\pm0.07}$ |
| | user-badge | $63.16_{\pm0.07}$ | $88.89_{\pm0.07}$ | $\mathbf{89.00}_{\pm0.05}$ | $\underline{88.93}_{\pm0.05}$ |
| rel-trial | study-outcome | $70.70_{\pm0.50}$ | $68.63_{\pm0.34}$ | $\underline{69.43}_{\pm0.46}$ | $\mathbf{70.92}_{\pm0.21}$ |

*Table 9.* Performance comparison on the **Entity Regression** task measured by MAE (↓). The best results are highlighted in **bold**, and the second-best results are underlined.

| Dataset | Task | Entity Mean | Entity Median | LightGBM | RDL | RelGNN | FROG |
|---------|------|-------------|---------------|----------|-----|--------|------|
| rel-avito | ad-ctr | 0.0462 | 0.0462 | $0.0408_{\pm0.0002}$ | $0.0413_{\pm0.0011}$ | $\underline{0.0393}_{\pm0.0008}$ | $\mathbf{0.0385}_{\pm0.0002}$ |
| rel-event | user-attendance | 0.3035 | 0.2692 | $0.2636_{\pm0.0002}$ | $0.2582_{\pm0.0058}$ | $\underline{0.2417}_{\pm0.0009}$ | $\mathbf{0.2408}_{\pm0.0022}$ |
| rel-fl | driver-position | 8.5013 | 8.5185 | $4.0685_{\pm0.0387}$ | $4.2074_{\pm0.1851}$ | $\underline{3.9807}_{\pm0.0841}$ | $\mathbf{3.8906}_{\pm0.0695}$ |
| rel-hm | item-sales | 0.1105 | 0.0780 | $0.0761_{\pm0.0000}$ | $\mathbf{0.0556}_{\pm0.0004}$ | $0.0564_{\pm0.0010}$ | $\underline{0.0561}_{\pm0.0001}$ |
| rel-stack | post-votes | 0.1061 | 0.0694 | $0.0679_{\pm0.0000}$ | $\mathbf{0.0652}_{\pm0.0001}$ | $0.0656_{\pm0.0002}$ | $\underline{0.0655}_{\pm0.0001}$ |
| rel-trial | study-adverse | 57.9303 | 57.9303 | $\mathbf{42.8452}_{\pm0.7798}$ | $46.1570_{\pm0.1783}$ | $46.6503_{\pm0.4006}$ | $\underline{44.7536}_{\pm0.9047}$ |
| | site-success | 0.4480 | 0.4411 | $0.4237_{\pm0.0020}$ | $0.4163_{\pm0.0082}$ | $\underline{0.3597}_{\pm0.0164}$ | $\mathbf{0.3504}_{\pm0.0222}$ |

*Table 10.* Performance comparison on the **Entity Recommendation** task measured by MAP (↑). The best results are highlighted in **bold**, and the second-best results are underlined.

| Dataset | Task | Global Popularity | Past Visit | LightGBM | RDL (GraphSAGE) | RDL (ID-GNN) | RelGNN | FROG |
|---------|------|-------------------|------------|----------|-----------------|--------------|--------|------|
| rel-avito | user-ad-visit | 0.00 | 1.95 | $0.05_{\pm0.00}$ | $0.04_{\pm0.01}$ | $\underline{3.73}_{\pm0.02}$ | $3.14_{\pm0.61}$ | $\mathbf{3.87}_{\pm0.04}$ |
| rel-hm | user-item-purchase | 0.30 | 0.89 | $0.34_{\pm0.03}$ | $0.82_{\pm0.07}$ | $\underline{2.78}_{\pm0.01}$ | $2.76_{\pm0.01}$ | $\mathbf{2.86}_{\pm0.01}$ |
| rel-stack | user-post-comment | 0.02 | 1.42 | $0.04_{\pm0.01}$ | $0.11_{\pm0.03}$ | $12.98_{\pm0.21}$ | $\underline{13.54}_{\pm0.18}$ | $\mathbf{13.58}_{\pm0.19}$ |
| | Post-post-related | 1.46 | 1.74 | $1.98_{\pm0.61}$ | $0.02_{\pm0.00}$ | $10.81_{\pm0.15}$ | $\underline{11.16}_{\pm0.00}$ | $\mathbf{11.48}_{\pm0.11}$ |
| rel-trial | condition-sponsor-run | 2.52 | 8.42 | $4.53_{\pm0.10}$ | $3.31_{\pm0.16}$ | $10.24_{\pm0.06}$ | $\underline{11.05}_{\pm0.10}$ | $\mathbf{11.20}_{\pm0.07}$ |
| | site-sponsor-run | 3.75 | 17.31 | $9.63_{\pm0.31}$ | $10.91_{\pm0.54}$ | $18.91_{\pm0.03}$ | $\mathbf{19.10}_{\pm0.09}$ | $\underline{19.09}_{\pm0.03}$ |

Table 11 shows that different datasets exhibit distinct preferences for relational representations. For example, on the driver-top3 task of rel-f1, modeling tables as edges has better performance, whereas on the user-repeat task of rel-event, modeling tables as nodes is clearly more effective.

Moreover, although the Random strategy does not consistently outperform the pure Table-as-Node or Table-as-Edge designs, our adaptive structure learning approach achieves the best overall performance. This result highlights the necessity of optimizing graph structures for effective learning.

*Table 11.* Ablation Study on the Roles of Table Modeling. The best results are highlighted in **bold**, and the second-best results are underlined.

| Metric | Dataset | Task | Table as Node | Table as Edge | Random | FROG |
|---|---|---|---|---|---|---|
| ROC-AUC (↑) | rel-fl | driver-top3 | $80.13_{\pm 0.17}$ | $\underline{80.62}_{\pm 0.91}$ | $79.83_{\pm 0.71}$ | $\mathbf{82.73}_{\pm 0.51}$ |
| | rel-event | user-repeat | $\underline{77.54}_{\pm 0.13}$ | $70.15_{\pm 0.30}$ | $76.08_{\pm 1.95}$ | $\mathbf{79.11}_{\pm 0.87}$ |
| MAE (↓) | rel-avito | ad-ctr | $0.0399_{\pm 0.0005}$ | $\underline{0.0390}_{\pm 0.0005}$ | $0.0391_{\pm 0.0002}$ | $\mathbf{0.0385}_{\pm 0.0002}$ |
| | rel-trial | site-success | $0.3959_{\pm 0.0191}$ | $0.4112_{\pm 0.0077}$ | $\underline{0.3655}_{\pm 0.0252}$ | $\mathbf{0.3504}_{\pm 0.0222}$ |
| MAP (↑) | rel-hm | user-item-purchase | $2.81_{\pm 0.01}$ | $\mathbf{2.87}_{\pm 0.01}$ | $2.83_{\pm 0.01}$ | $\underline{2.86}_{\pm 0.01}$ |
| | rel-stack | post-post-related | $10.76_{\pm 0.13}$ | $10.77_{\pm 0.27}$ | $\underline{11.04}_{\pm 0.45}$ | $\mathbf{11.48}_{\pm 0.11}$ |

## F.4. Sensitivity Analysis of $\beta, \gamma$

To evaluate the robustness of the proposed framework, we conduct a sensitivity analysis on the hyperparameters $\beta$ and $\gamma$, which govern the contributions of the $\mathcal{L}_{emb}$) and $\mathcal{L}_{pair}$, respectively.

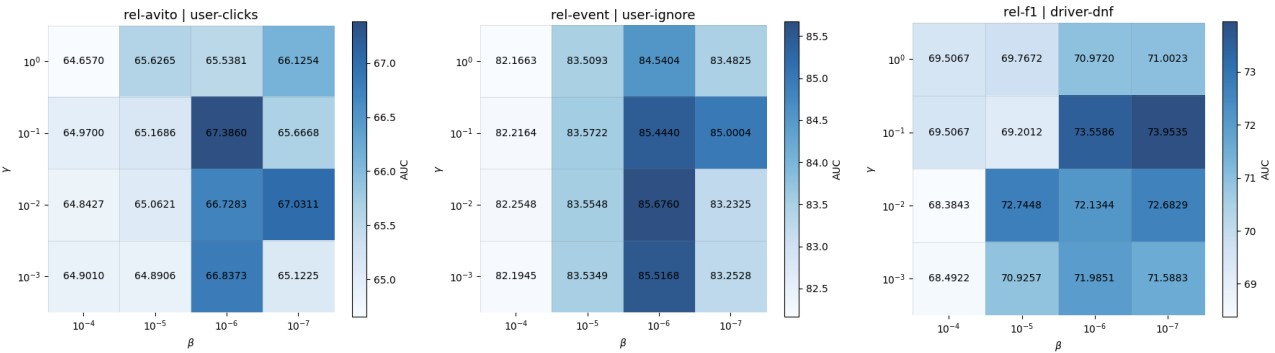

*Figure 10.* Hyperparameter Sensitivity Analysis of $\beta, \gamma$

As illustrated in Fig. 10, both $\mathcal{L}_{emb}$ and $\mathcal{L}_{pair}$ exhibit distinct unimodal peak behaviors across varying ranges of $\beta$ and $\gamma$. An observation is the consistency of the effective hyperparameter intervals across diverse datasets. Specifically, peak performance is consistently achieved near $\beta \approx 10^{-6}$ and $\gamma \approx 0.1$, which are the values used in our experiments. This cross-dataset stability suggests that the properties captured by $\mathcal{L}_{FD}$ are broadly useful.

## F.5. Ablation Study: Role of $\mathcal{L}_{FD}$

To investigate the effect of $\mathcal{L}_{FD}$, we conduct ablation studies by removing $\mathcal{L}_{emb}$ and $\mathcal{L}_{pair}$.

The results in Table 12 show performance degradation to varying degrees. The results indicate that both $\mathcal{L}_{emb}$ and $\mathcal{L}_{pair}$ have a significant impact, demonstrating the necessity of the FD constraint.

## F.6. Schema Graphs of the RDBs used in Section 5.4

To better evaluate the model's ability to capture relational information, Section 5.4 visualizes results on two datasets. The rel-event dataset contains multi-relations, while rel-avito exhibits richer relational structures among tables. These

*Table 12.* Ablation study on removing $\mathcal{L}emb$ and $\mathcal{L}pair$.

| Task | FROG | No $\mathcal{L}_{emb}$ | No $\mathcal{L}_{pair}$ |
|---|---|---|---|
| rel-avito \| user-clicks (AUC↑) | **67.25** | 64.43 (↓2.8) | 65.49 (↓1.8) |
| rel-event \| user-ignore (AUC↑) | 85.43 | 78.75 (↓6.7) | **85.69** (↑0.3) |
| rel-f1 \| driver-dnf (AUC↑) | **74.39** | 73.25 (↓1.1) | 72.23 (↓2.2) |
| rel-trial \| site-success (MAE↓) | **0.3564** | 0.4012 (↑0.04) | 0.3744 (↑0.02) |
| rel-stack \| post-post-related (MAP↑) | **11.44** | 10.90 (↓0.5) | 9.92 (↓1.5) |
| rel-hm \| user-item-purchase (MAP↑) | **2.87** | **2.87** (↓0.00) | 2.86 (↓0.01) |

characteristics make the two datasets well suited for assessing FROG 's relational learning capability.

Their RDB relationships are illustrated in Fig. 11 (Robinson et al., 2024):

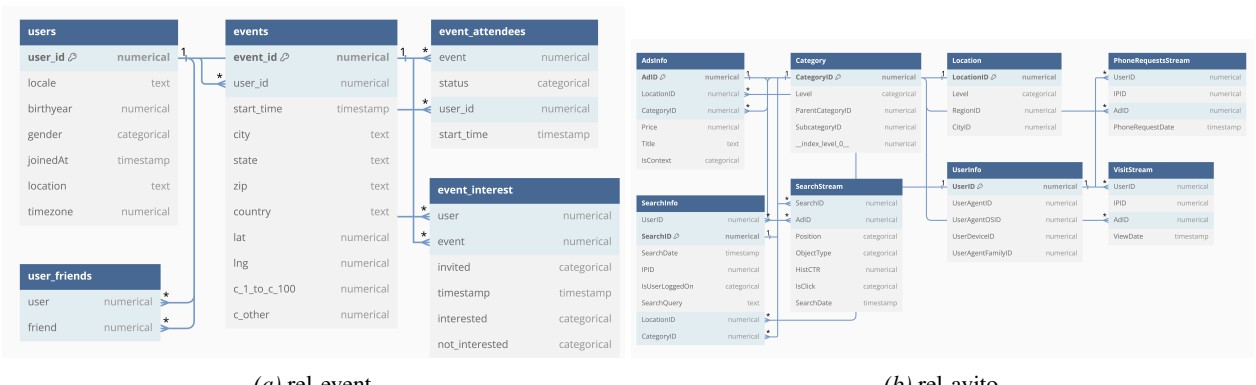

*(a)* rel-event                                                      *(b)* rel-avito

*Figure 11.* Visualization of rel-event and rel-avito

The corresponding schema graphs are shown in Fig. 12.

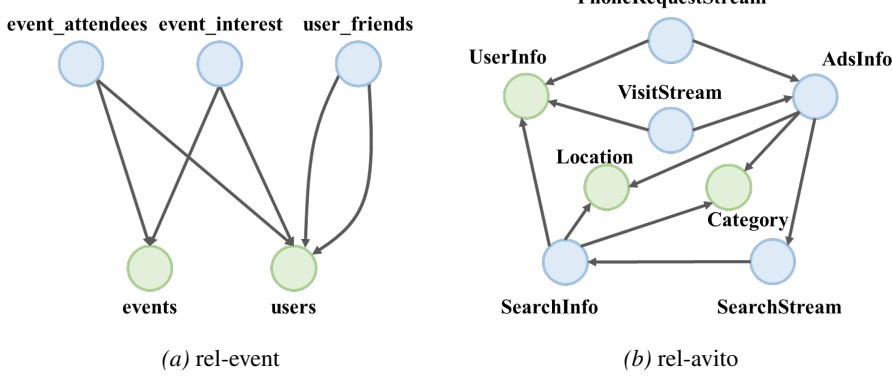

*(a)* rel-event                                  *(b)* rel-avito

*Figure 12.* Schema Graphs of rel-event and rel-avito

## F.7. Node or Edge? Analysis of Learned Graph Structures

Our FROG not only produces a trained model after training, but also obtains the optimized graph structure of the Schema graph. To investigate the roles of tables under different relations, we collected the following statistics for each dataset:

- The modeling roles of tables to which intermediate nodes belong in the Co-Occurrence paradigm.

- Whether the modeling roles of tables for intermediate nodes change across information propagation in two directions within the Co-Occurrence paradigm (`same` indicates consistent roles, `diff` indicates inconsistent roles).

- The modeling roles of tables to which intermediate nodes belong in the Completion paradigm.

The results are shown in Fig. 13:

- In the Co-Occurrence paradigm, intermediate nodes tend to be modeled as nodes for classification and regression tasks, while they are more naturally modeled as edges for link prediction tasks.

- In the Co-Occurrence paradigm, across different directions of information propagation, tables tend to maintain *consistent roles*.

- In the Completion paradigm, the modeling preferences of tables vary across datasets, and the tendencies are generally more pronounced than in the Co-Occurrence paradigm. For instance, rel-trial favors table-as-node, while rel-event favors table-as-edge. The rel-hm dataset has been omitted here because such a paradigm does not exist in this dataset.

## G. Hardware and Software Configurations

We conduct the experiments with:

- Operating System: Ubuntu 22.04.4 LTS.
- CPU: Intel(R) Xeon(R) Silver 4110 CPU @ 2.10GHz.
- GPU: NVIDIA Tesla V100 SMX2 with 32GB of Memory.

## H. Limitation and Future Work

**Limitaion.** Compared with GNN-based methods, FROG has higher computational cost, mainly due to computing table-as-edge induced two-hop paths and increased parameter size. This overhead brings benefits on multi-hop tasks such as user-clicks of rel-avito (*i.e.* , User $\rightarrow$ SearchInfo $\rightarrow$ SearchStream $\rightarrow$ Ad), but on simpler datasets like rel-hm (article $\leftarrow$ transactions $\rightarrow$ customer) the gains are limited.

**Future work.** First, one possible improvement is to precompute candidate table-as-edge relations to avoid computation during training, thereby improving training efficiency. In addition, future work could explore adaptive mechanisms to apply table role modeling only when it is beneficial, in order to reduce unnecessary overhead on simpler tasks.

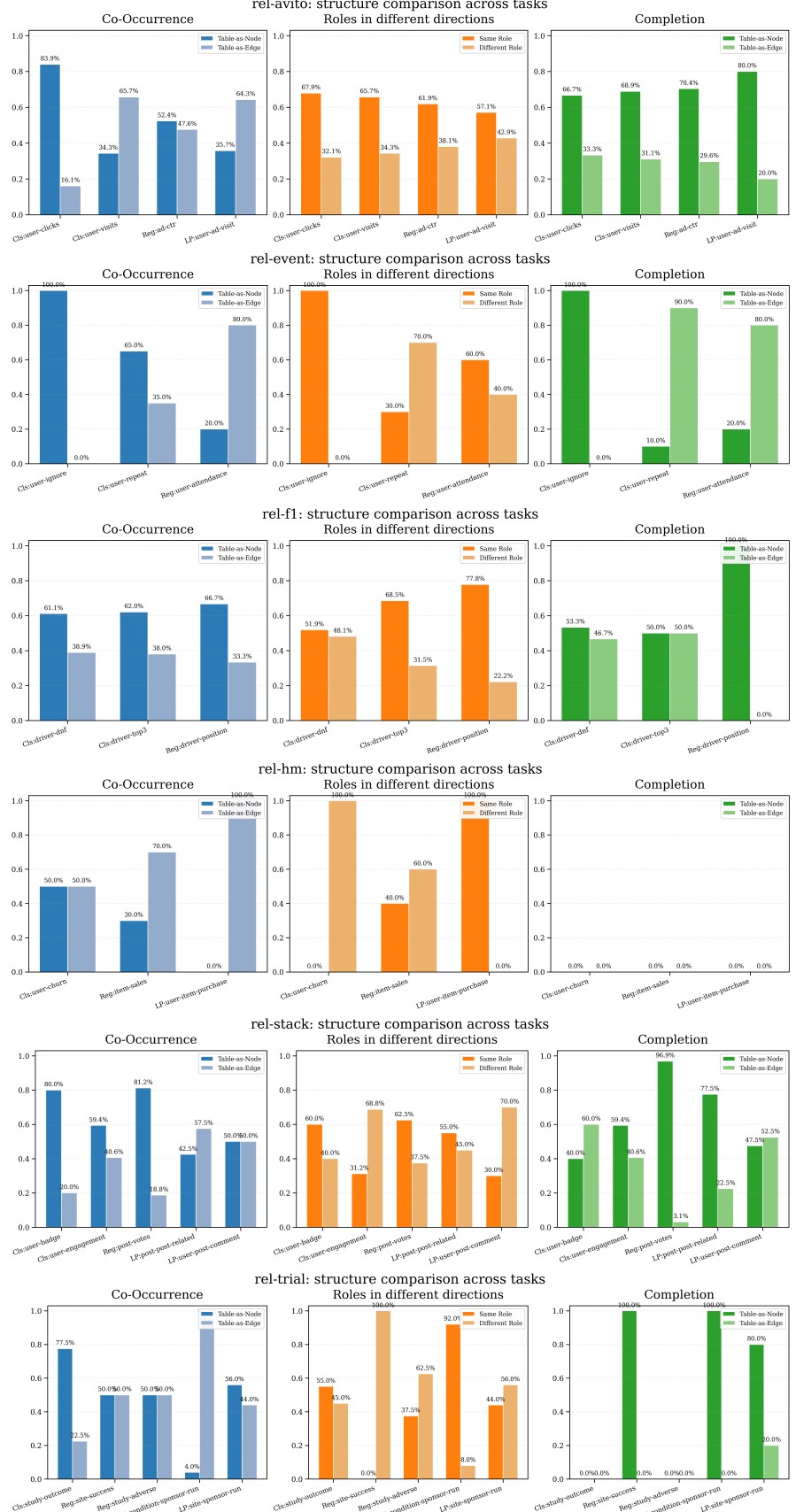

*Figure 13.* Statistical analysis of the modeling methods of the table under different relationships

