# OpenReview forum: "Is Fixing Schema Graphs Necessary? Full-Resolution Graph Structure Learning for Relational Deep Learning"
_ICML.cc/2026/Conference — ICML 2026 regular_

### Official Review · Reviewer_ozyD · 2026-03-02

**Soundness:** 2
**Presentation:** 2
**Significance:** 3
**Originality:** 2
**Overall Recommendation:** 3
**Confidence:** 3

**Summary:**

This paper studies graph structure learning for Relational Deep Learning (RDL) on relational databases. Authors claim that constructing a fixed schema graph over a relational database before applying a GNN, as done by RDL methods, limits flexibility during message passing. However, modifying the graph structure is non-trivial because RDL must satisfy the full-resolution property, meaning that the constructed graph must contain the full information of the original database.
To address this, the authors propose FROG, a framework that optimizes table roles, allowing tables to contribute as nodes and edges in message passing. They prove that both role assignments preserve full-resolution and introduce relation-driven message passing together with functional dependency (FD) regularization to maintain relational consistency.
Experiments show that learning table roles improves downstream performance compared to fixed schema constructions.

**Compliance With Llm Reviewing Policy:**

Affirmed.

**Key Questions For Authors:**

- Under what conditions is the table-as-node vs. table-as-edge assignment uniquely identifiable? Could multiple assignments yield equivalent REG representations?
- How sensitive are results to the weighting of $L_{FD}$  ? Is there a risk of over-regularization harming predictive performance?

- If a learned schema graph is optimized for one downstream task, does it transfer to another task on the same RDB?

- Could continuous edge reweighting (while preserving full-resolution) provide a simpler alternative?

**Limitations:**

The paper focuses primarily on methodological and technical aspects of relational graph structure learning, and does not provide a substantial discussion of limitations. It would be beneficial to add a discussion including:
- Scalability and Computational Overhead
- Stability of Learned Schema Roles
- Task-Specific Overfitting of Schema Graphs

The paper does not explicitly discuss potential negative societal impact of their work

**Strengths And Weaknesses:**

**Strenghts**

-  The problem identified in the paper is non-trivial in RDL approaches: the need to preserve relational semantics (full-resolution) conflicts with standard GSL techniques that alter edges.
- The formalization related to information theory ($H(X∣G)=0$) provides a foundation to discuss the problem from another perspective.
- Instead of learning adjacency entries, the method learns table roles (node vs. edge). This is an elegant workaround to the full-resolution restriction and reframes GSL in RDL in a principled way.
- Introducing FD losses at table and entity levels is conceptually meaningful and tightly connected to database theory. This strengthens the semantic grounding of learned representations.

**Weaknesses**

- While the paper provides a clean information-theoretic formulation of the full-resolution property and proves which operations violate it the theoretical analysis does not extend to the optimization of table roles itself. In particular:
   - The space of admissible schema configurations under full-resolution is not formally characterized.
   - There are no guarantees regarding identifiability or uniqueness of learned table roles.
   - The claim that table-as-edge better captures long-range dependencies is motivated intuitively (via mutual information arguments), but lacks a formal guarantee.
- While the paper formulates schema optimization as a constrained structural learning problem, (eq. 4) in practice the optimization appears to be mediated indirectly through message-passing parameterization and attention mechanisms. The paper would benefit from a clearer formal characterization of the structural search space and the nature (discrete vs continuous) of the learned schema.
- The proposed framework introduces additional components (i.e. role learning, FD regularization, alternating optimization), but the paper does not provide a quantitative comparison of computational complexity (e.g., training time, memory usage) against baseline RDL models. Given that the reported performance gains are relatively modest in several settings (Tables 1, 2, 3), it remains unclear whether the additional complexity is justified in practice

---

> ### Author Rebuttal · Authors · 2026-03-31
>
> Thank you for your valuable feedback. Below are our responses to your questions.
>
> 1. **Search Space and the Nature of the Learned Schema.** From the perspective of problem modeling, this can be viewed as searching for table roles in a discrete space of size $2^{|T|}$. Considering that tables may take different roles across relations, the space further expands to $2^{|rel_E|}$, where $|rel_E|$ is the number of 2-hop relations. However, such a large search space is impractical in practice due to its high complexity. Therefore, **we consider a more fundamental perspective: the roles of different tables essentially correspond to how a GNN acquires information during message passing**. Based on this, we define message passing patterns and gating mechanisms, transforming the discrete selection into a continuous end-to-end optimization over representations, searching only a space on the order of $|rel_E|$. Thus, the original selection problem is a special case of the continuous formulation. We will clarify this distinction and the relaxed search space in the revision.
>
> 2. **Complexity.** FROG introduces a moderate computational overhead compared to RelGNN. However, inference efficiency can be reduced by approximately 20% by precomputing global 2-hop relations prior to training. The model increases the number of parameters by less than 20% on average to enable edge convolution and gating. For details, please refer to our responses to Reviewer xUwP (#1 and #2).
>
> 3. **Uniqueness of REG.** Since the learned table roles are continuous values, we first discretize them using 0.5 as the threshold (<0.5 is node, > 0.5 is edge). Under the setting of 23 tasks with five runs per task, the average probability that each table maintains a consistent role in 2-hop (i.e., the role with higher probability, acting as either a node or an edge) is **87.23%**, and **57.35%** of the tables remain in the **same role across all five runs**. We believe that such consistency reflects the role of a table within a relation. For example, in rel-hm (three tables: article–transaction–customer), the transaction table is **100% identified as an edge** in the user-item-purchase task, since treating it as an edge shortens the information path between customer and article, helping the model better capture their relationships.
>
> 4. **Sensitive of $L_{FD}$.** From the experimental results (see Figure [here](https://anonymous.4open.science/r/FORG-5805/rebuttal/combined.png)), both $L_{\mathrm{emb}}$ and $L_{\mathrm{pair}}$ (components of $L_{\mathrm{FD}}$), exhibit peak behaviors as $\beta$ and $\gamma$ vary, demonstrating the effectiveness of the regularization. Moreover, we observe that the effective hyperparameter ranges are similar across different datasets ($\beta\approx 1e^{-6}, \gamma\approx 0.1$). Therefore, we fix $\beta$ and $\gamma$, which indicates the generalizability of $L_{\mathrm{FD}}$ and shows that the current setting avoids the risk of over-regularization.
>
> 5. **Transferability of the Structure.** To study the role of tables in cross-task settings, we conducted experiments on a unified dataset by training each task with structures learned from other tasks. We show the results on rel-event (simpler structure) and rel-avito (covering three task types). The results (the i-th row and j-th column represents the result of the i-th task trained under the structure learned from the j-th task, best in **bold**, second-best in *italics*) show that table roles **can generalize across tasks and achieve competitive performance**, with some cross-task structures even outperforming task-specific ones. However, in most cases, the best results are still obtained using structures learned from the target task, likely because different tasks emphasize table relationships differently.
>
>    | rel-event | user-repeat | user-ignore | user-attendance |
>    | - | - | - | - |
>    | **user-repeat** (AUC↑)| **79.01**| *76.30*| 74.31|
>    | **user-ignore** (AUC↑)| 83.32| **84.59**| *84.39*|
>    | **user-attendance** (MAE↓) | *0.263534*| 0.263535| **0.243501**|
>
>    | rel-avito| user-visits | user-clicks | ad-ctr   | user-ad-visit |
>    | - | -- | ----------- | -------- | ------------- |
>    | **user-visits** (AUC↑)| *66.37*| 66.18| 66.36| **66.54**|
>    | **user-clicks** (AUC↑)| 65.58| **67.15**| 65.52| *66.03*|
>    | **ad-ctr** (MAE↓)| 0.0390| **0.0380**| *0.0385*| 0.0386|
>    | **user-ad-visit** (MAP↑)| 3.84| **3.88**| 3.86| *3.87*|
>
> 6. **Continuous Edge Reweighting.** In fact, FROG uses continuous edge reweighting, i.e., moving from discrete selection of tables as nodes/edges to continuous values. This approach not only preserves full-resolution, but also enables the model to more deeply explore associations. Figure 7 in paper demonstrates the superiority of this continuous formulation.
>
> 7. **Lack of Explanation.** In response to the missing and insufficient content regarding the problem definition and search space, we will further formalize these aspects in the paper.

---

> > ### Author Rebuttal · Reviewer_ozyD · 2026-04-03
> >
> > Thank you for the thorough rebuttal. Most of my concerns have been adequately addressed, in particular the search space clarification, the computational overhead, the hyperparameter sensitivity, and the transferability experiments.
> >
> > One point that remains unaddressed is the performance on tasks where FROG does not clearly outperform RelGNN (e.g., rel-hm user-churn, rel-stack user-engagement/user-badge). A brief discussion of when the learned structure is expected to help would strengthen the paper.

---

> > > ### Author Response · Authors · 2026-04-05
> > >
> > > Through further analysis, we observe that the effectiveness of FROG depends on the nature of relational dependencies required by the task.
> > >
> > > In scenarios where prediction relies on multi-hop relational paths, FROG provides clear benefits. For example, in rel-avito (user-clicks task), the relevant information follows a multi-hop chain (User → SearchInfo → SearchStream → Ad), where intermediate tables encode rich interaction signals. In such cases, modeling these relations as edges enables more direct information flow and reduces repeated aggregation over intermediate nodes.
> > >
> > > In contrast, datasets such as rel-stack and rel-hm involve relatively shallow relational dependencies. For instance, in rel-stack, most task-relevant signals for user-engagement can be obtained within 1-hop relations (e.g., user-post, user-vote, user-comment). Similarly, rel-hm contains a simple structure (3 tables: article-transaction-customer), where aggregation of RelGNN is sufficient. In these cases, the advantage of structure learning becomes less pronounced, and the additional flexibility may not translate into significant gains. Notably, however, the performance gap between FROG and RelGNN remains consistently small (on average <0.5%, with the minimum <0.1%).
> > >
> > > We further note that FROG is not solely dependent on learned structure. The FD regularization, particularly the embedding constraint $L_{emb}$, plays an important role. For example, on rel-event, despite its relatively simple schema, enforcing structured embedding space leads to significant improvements (over 5% compared to removing $L_{emb}$).
> > >
> > > Overall, FROG is most beneficial for RDL tasks that require long-range relational information, where table-as-node/edge design helps preserve and propagate information more effectively.
> > >
> > > We sincerely thank you again for your constructive feedback and hope that these additional explanations help address your remaining concerns. We would greatly appreciate your consideration in revising the evaluation in light of these clarifications.

---

### Official Review · Reviewer_G4rZ · 2026-03-10

**Soundness:** 4
**Presentation:** 3
**Significance:** 4
**Originality:** 4
**Overall Recommendation:** 6
**Confidence:** 4

**Summary:**

This paper studies graph construction for relational deep learning on relational databases. Instead of using a fixed schema graph, the authors propose FROG, a framework that performs full-resolution graph structure learning by modeling relational tables as either nodes or edges and learning their roles during message passing. The approach also introduces functional dependency constraints to preserve relational semantics during representation learning. Experiments on multiple RelBench datasets demonstrate consistent improvements across classification, recommendation, and regression tasks.

**Compliance With Llm Reviewing Policy:**

Affirmed.

**Final Justification:**

The rebuttal addressed most of my concerns. The additional analysis and ablations make the paper much more convincing. I view this work as a very insightful and interesting direction for relational deep learning, since it moves beyond fixed schema graphs toward full-resolution graph structure learning for relational databases. Overall, I am inclined to raise my score to show my support for acceptance.

**Key Questions For Authors:**

(1) If the same relational database is used for different downstream tasks, do the learned table roles remain consistent, or does the model learn significantly different schema structures?

(2) Can the learned table-as-node/edge decisions be used to produce interpretable schema graphs that reveal new insights about the relational dataset?

**Limitations:**

yes

**Strengths And Weaknesses:**

Strengths:

(1) The paper identifies an important limitation in current RDL pipelines: the schema graph is typically fixed before GNN training, which may constrain representation learning.

(2) The work introduces the idea that table roles can be learned rather than predefined, providing a new perspective on graph construction for relational data.

(3) The paper formalizes the full-resolution property using an information-theoretic formulation $H(X∣G)=0$, which provides a principled constraint for graph structure learning in relational databases.

Weaknesses:

(1) The analysis of experimental results could be deeper. The paper only analyzed the statistical results of "table" as different roles, but did not provide any further insights on how "table-as-node/edge" would be better in certain situations.

(2) The method introduces several modules simultaneously (role gating, two message passing paradigms, FD loss, alternating optimization). It is somewhat difficult to isolate which component contributes most to the final gains.

(3) Relational databases can be extremely large, but the paper does not thoroughly analyze computational cost or memory overhead compared to standard RDL pipelines.

---

> ### Author Rebuttal · Authors · 2026-03-31
>
> Thank you for your valuable feedback. Below are our responses to your questions.
>
> 1. **Deeper Analysis of Table Roles.** We analyze the table roles learned by the model across 23 tasks and visualize them (see figure [here](https://anonymous.4open.science/r/FORG-5805/rebuttal/datasets_vertical_combined.png)). Overall, an intuitive observation is that behaviors within the same dataset show certain similarities, such as rel-avito tending toward table-as-node, while rel-hm tends toward table-as-edge.
>
>    From a more detailed perspective, we believe that the **role of a table intuitively reflects its importance in the current relation**. For example, in rel-avito, where the task is to predict a user’s future ad views, the user table is directly connected to the searchinfo, visitstream, and phonerequest tables. These three tables **already provide rich user behavior information** from aspects like search actions and temporal frequency, whereas its 2-hop neighbor, the AdsInfo table, is less important (since it **only** contains price and title information). In this case, table-as-node can extract more information. In the experiment, it was indeed true that these three tables were modeled as nodes with a probability of over 90%.
>
>    In contrast, for rel-hm, there are only three tables: customer-transaction-article, aiming to capture customer purchase behavior. Here, the transaction table contains **only** price, sale channel, and time information, which is far from sufficient to characterize customer behavior, whereas the article table has **over a dozen fine-grained features**. Therefore, in this scenario, modeling transaction as an edge is more suitable, serving as a **bridge** for the customer to directly access richer and more useful information, which is consistent with the experimental results.
>
> 2. **Component importance.** We conduct ablation studies by removing $L_{\mathrm{emb}}$, $L_{\mathrm{pair}}$, and the table-as-edge component. The results show performance degradation to varying degrees. In addition, Figure 7 in the paper provides supplementary evidence. The results indicate that $L_{\mathrm{emb}}$, have the most significant impact. Similarly, $L_{\text{pair}}$ also has a positive effect in most cases. These results demonstrate the necessity of the FD constraint. Additionally, table-as-edge also plays an important role, especially in the entity recommendation task. The joint design of the loss and the message passing paradigm leads to the final performance.
>
>    | Task | FULL | No $L_{emb}$  | No $L_{pair}$ | No table-as-edge |
>    | - | - | - | - | - |
>    | rel-avito \| user-clicks (AUC↑)   | **67.25**  | 64.43 (↓2.8)     | 65.49 (↓1.8)     | 66.48 (↓0.8)     |
>    | rel-event \| user-ignore (AUC↑)    | 85.43      | 78.75 (↓6.7)     | **85.69** (↑0.3) | 85.62 (↑0.2)     |
>    | rel-f1 \| driver-dnf (AUC↑)     | **74.39**  | 73.25 (↓1.1)     | 72.23 (↓2.2)     | 72.64 (↓1.7)     |
>    | rel-trial \| site-success  (MAE↓)    | **0.3564** | 0.4012 (↑0.04)   | 0.3744 (↑0.02)   | 0.4066 (↑0.05)   |
>    | rel-stack \| post-post-related  (MAP↑) | **11.44**  | 10.90 (↓0.5)     | 9.92 (↓1.5)      | 10.76 (↓0.7)     |
>    | rel-hm \| user-item-purchase  (MAP↑)   | **2.87**   | **2.87** (↓0.00) | 2.86 (↓0.01)     | 2.81 (↓0.06)     |
>
> 3. **Complexity.** FROG introduces a moderate computational overhead compared to RelGNN. However, inference efficiency can be reduced by approximately 20% by precomputing global 2-hop relations prior to training. The model increases the number of parameters by less than 20% on average to enable edge convolution and gating. For details, please refer to our responses to Reviewer xUwP (#1 and #2).
>
> 4. **Uniqueness of REG.** The learned table roles show high consistency, with each table maintaining its role with an average of **87.23%** consistency across 23 tasks (five runs each), and **57.35%** of tables maintaining the same role across all five runs. For details, please refer to our responses to Reviewer ozyD (\#3).
>
> 5. **Interpretable Schema Graph.** We thank the reviewers for their valuable suggestions. Indeed, we believe that the learned table roles can be used to produce interpretable schema graphs. As described in #1 Deeper Analysis of Table Roles, we linked the number of effective features a table contains for the target task with the table roles learned by the model, enabling an interpretable analysis of the data. Experimental results also validate our hypothesis.
>
>    Furthermore, by examining the roles of tables in different relations, we can analyze the importance of table information for the task from the perspective of information flow paths. For example, in the F1 dataset, in **drivers-results-constructors**, results takes the role of an **edge** because a driver reflects the strength of their constructor team; whereas in **races-results-drivers**, results serves as a node because, for the current task, we care about the driver’s final results rather than the specific races.

---

> > ### Author Rebuttal · Reviewer_G4rZ · 2026-04-04
> >
> > The rebuttal addressed most of my concerns. The additional analysis and ablations make the paper much more convincing. I view this work as a very insightful and interesting direction for relational deep learning, since it moves beyond fixed schema graphs toward full-resolution graph structure learning for relational databases. Overall, I am inclined to raise my score to show my support for acceptance.

---

### Official Review · Reviewer_1rQv · 2026-03-12

**Soundness:** 2
**Presentation:** 1
**Significance:** 1
**Originality:** 2
**Overall Recommendation:** 2
**Confidence:** 4

**Summary:**

This paper proposes FROG, a Full-Resolution and Optimizable Graph Structure Learning framework for Relational Deep Learning (RDL). The key idea is to allow relational tables to serve as either nodes or edges in the Relational Entity Graph (REG), with these roles learned adaptively via a gating mechanism rather than fixed by the schema graph. The paper defines a "full-resolution" property, requiring that the original RDB is losslessly recoverable from the constructed graph, and argues that standard graph structure learning (edge pruning/addition) violates this property. The authors further prove that their table-as-node and table-as-edge formulations preserve full-resolution, and that table-as-edge preserves strictly more mutual information than table-as-node under certain assumptions. To enforce functional dependency (FD) constraints in learned representations, a regularization loss is introduced. Experiments on 6 datasets and 23 tasks from the RelBench benchmark show competitive performance against LightGBM, GraphSAGE, ID-GNN, and RelGNN baselines.

**Compliance With Llm Reviewing Policy:**

Affirmed.

**Final Justification:**

My final recommendation is to reject this paper in its current form. The rebuttal and the follow-up comment did not resolve the core concerns from my review, and the authors' own comments confirm several of them.

**Soundness.** The central theoretical contribution is the full-resolution property and its associated theorems. In their additional responses, the authors agree that these results are "better interpreted as design principles rather than strict theoretical necessity." This is precisely what my review argued: the theory motivates the design but does not provide actionable guarantees about downstream performance. Further, Theorems 4.1 and 4.2 are trivial, and this fact was not discussed in either response.

**Originality.** The idea of learning table roles (node vs edge) is reasonable and worth exploring. However, the paper frames it as a novel form of graph structure learning and supports it with theoretical machinery that adds little beyond what is obvious. The practical insight is buried under unnecessary formalism.

**Significance.** The empirical improvements over RelGNN are marginal and often within overlapping standard deviations, while the method introduces substantial architectural complexity (per-relation gating, two message passing modules, FD regularization with two losses, alternating optimization). The authors' response that only about three cases show overlapping results does not address the broader concern: the ratio of added complexity to performance gain is questionable.

**Clarity.** The authors acknowledged that excessive formalism hinders readability and promised revision. However, the issues are structural, not cosmetic: standard operations are named as novel paradigms, and the theoretical sections need to be reframed or even removed rather than just rephrased.

**Impact of the rebuttal.** The rebuttal partially addressed the hyperparameter concern, which I accepted. The remaining core issues, such as the theory-practice disconnect (confirmed by the authors themselves), the triviality of the theorems (never discussed), and marginal empirical gains relative to complexity, were not resolved. The follow-up comment conceded that the theory is design motivation rather than formal necessity, which aligns with my original critique and reinforces the need for major revision. My assessment remains unchanged.

**Suggestions for revision.** The core idea of learning table roles is sound and potentially useful. I encourage the authors to reframe the theoretical sections as design intuition, simplify the presentation by removing paradigm names for standard operations, and provide stronger empirical separation from baselines. A substantially revised version could become a solid contribution at a future venue.

**Key Questions For Authors:**

1. Can you provide a concrete example (or formal argument) where violating the full-resolution property leads to degraded downstream task performance? Without such evidence, why should a practitioner prefer a full-resolution method over a standard one that achieves better results?

2. Propositions 3.3 and 3.4 show that modified REGs cannot recover the original RDB, but a pruned/augmented REG is still a valid REG for some RDB instance. What specific harm results from this? Is there a scenario where edge pruning or addition produces a graph that violates schema constraints (not just changes the implied data instance)?

3. In what precise sense is choosing between table-as-node and table-as-edge a form of graph structure learning? Standard GSL modifies edges in a graph; here, the operation is a per-table binary architectural choice that does not add or remove any edges. Could the authors clarify the connection to GSL beyond a high-level analogy?

4. In Theorem 4.2, the non-information-preserving assumption for the intermediate node aggregation in the table-as-node strategy is central. Have you empirically verified that this assumption holds for the specific GNN architectures and datasets used in your experiments? If the first-hop aggregation happens to be nearly information-preserving, the advantage of table-as-edge should vanish.

5. The search space in Table 5 lists only standard GNN hyperparameters and omits FROG-specific ones ($\beta$, $\gamma$, $d$, $\alpha$). Was the same search applied to all the considered baselines? FROG has strictly more tunable hyperparameters than RelGNN or RDL, and the performance gaps are often within 1-2 points with overlapping standard deviations. How can we distinguish a methodological improvement from a larger effective hyperparameter budget?

**Limitations:**

The paper does not include an explicit limitations section. The most significant unacknowledged limitation is the disconnection between the theoretical framework (concerning lossless data reconstruction) and the practical objective (downstream task performance). The paper also does not discuss computational overhead of the alternating optimization procedure (Algorithm 2) relative to standard RDL pipelines, or potential failure modes of the adaptive table role assignment. Societal impact is not a concern for this work.

**Strengths And Weaknesses:**

## Strengths

1. **Relevant problem.** The question of whether schema graphs should be fixed or jointly optimized in RDL is timely and practically important. The observation that table roles (node vs edge) can affect downstream performance (Figure 7, Table 9) is a useful direction to explore for the RDL community.

2. **Learnable table role assignment.** The idea of learning whether a table should be a node or an edge (Equations 6-8) is reasonable, and framing it as a role selection problem rather than edge manipulation is an interesting perspective.

3. **Comprehensive experimental evaluation.** The evaluation covers 6 datasets, 23 tasks, and 3 task types (classification, regression, recommendation) from RelBench, with proper trial averaging and standard deviations. The ablation studies on table roles (Section 5.3) and FD constraints (Section 5.4) are informative.

## Weaknesses

The paper brings a recurring pattern across Sections 3.3-4.4: theoretical statements and formal notation are used to dress up ideas that are either trivially true, practically irrelevant, or much simpler than their presentation suggests. The concerns are detailed below.

1. **Section 3.3: The central theoretical motivation is disconnected from the practical objective.**

- The full-resolution property is about lossless data reconstruction rather than downstream predictive performance, and these are fundamentally different objectives. In machine learning, we routinely and successfully employ lossy transformations, such as dimensionality reduction, pooling, dropout, feature selection, all of which break full-resolution by design and yet improve performance. The paper provides no formal argument (e.g., a lower bound on task error as a function of information loss, or some negative result for non-full-resolution methods) connecting full-resolution to downstream performance. Without such a connection, the requirement can remain a reasonable inductive bias, but not a theoretically grounded necessity.
- Propositions 3.3 and 3.4 formalize a trivially true and practically uninteresting fact. Both state that modifying the REG makes it impossible to recover the *original* RDB. However, an RDB schema admits infinitely many valid instantiations — any modified REG is still a valid REG that corresponds to *some* RDB instance. Pruning an edge yields a graph that is full-resolution with respect to a smaller database; adding an edge yields one for a larger database. These propositions therefore can be reduced to the phrase "changing a data structure means that you can no longer recover the original" — true, but with no practical implication for whether the modified graph is useful for prediction. The paper would benefit from either showing that specific *schema constraint violations* arise from GSL operations or reframing these results as design motivation rather than theoretical necessity.

2. **Section 4.1: Confusing presentation of a simple idea with a trivial theorem.**

- The section merges the GSL motivation, the table role selection idea, and the full-resolution analysis in a way that obscures what is actually being proposed. It claims to "reinterpret the GSL objective" as determining table roles, but this is not graph structure learning in any standard sense — standard GSL learns a continuous or discrete adjacency matrix, while here the method makes a binary architectural choice (node or edge) per table. Calling this GSL can be misleading for readers.
- Theorem 4.1 is correct but trivially true. It states that if each table is mapped via either $f_n$ (table-as-node) or $f_e$ (table-as-edge), both individually invertible (Lemmas B.4 and B.5) and applied to disjoint data, their combination is also invertible. The mention of a "self-learning decision function" $g$ suggests the learnability matters, but *any* $g$ (random, constant, or learned) preserves full-resolution. The theorem just formalizes what is obvious from Figure 5: reformatting data without discarding anything is lossless.
- The actually interesting question about which table role assignment leads to better *predictions* is not addressed by the theory at all and is left entirely for the empirical evaluation (Section 5.3, Table 9).

3. **Section 4.2: The presented theorem is a direct application of the DPI under a strong assumption.**

- Theorem 4.2 states that table-as-edge preserves more mutual information about the source entity than table-as-node. This follows almost immediately from the DPI applied to the Markov chain from Lemma C.1 — the critical assumption that the first-hop aggregation $f_n^1$ is not information-preserving does all the work. Under this assumption, the conclusion (two lossy steps lose more than one direct step) is expected and not surprising.
- It compares supremums over function classes $\mathcal{F}_N$ and $\mathcal{F}_E$ under the assumption of "sufficiently large" embedding dimension, but the practical relevance depends on whether these supremums are reached by the actual neural networks, which is not analyzed.

4. **Section 4.3: Standard neural network operations presented with heavy notation and terminology.**

- The two "relation-driven message passing paradigms" reduce to simple operations: Co-Occurrence (Equation 10) is concatenation of three embeddings followed by a linear layer, and Completion (Equation 11) adds a sigmoid gating mechanism on top of the same concatenation. These are standard building blocks that do not need special paradigm names or elaborate notations.
- The section raises the complication that table-as-edge "may induce hyperedges among multiple tables" only to immediately resolve it by decomposing into standard edges, without formal explanation of why this decomposition preserves the information.
- For the table-as-node case, the section acknowledges that "standard heterogeneous GNNs can be directly applied" (line 266). Thus, the entire section could be condensed to a single paragraph describing the table-as-edge message computation without loss of clarity or content.

5. **Marginal empirical improvements over GNNs do not justify the added complexity.** Examining Tables 1-3, the performance gap between LightGBM and GNN-based methods is often dramatic and meaningful (e.g., `user-visits` classification: 52.82 to 66.42 ROC-AUC; `user-post-comment` recommendation: 0.04 to 13.58 MAP). In contrast, the gap between FROG and RelGNN is consistently small and frequently within overlapping standard deviations. In classification (Table 1), gains range from +0.52 on `driver-top3` to +2.94 on `user-ignore`, with most under 1 point. In recommendation (Table 2), several gains are negligible: +0.04 on `user-post-comment`, +0.15 on `condition-sponsor-run`. In regression (Table 3), FROG does not consistently win: RDL beats both RelGNN and FROG on `item-sales` and `post-votes`, and LightGBM outperforms all GNN methods on `study-adverse`. These marginal and inconsistent improvements come at the cost of substantial architectural complexity: learnable per-relation gating, two message passing modules, a special GNN per relation type, FD regularization with two losses and a separate scoring model $\mathcal{M}^S$, alternating optimization with frozen/unfrozen phases (Algorithm 2), and additional hyperparameters ($\beta$, $\gamma$, low-rank dimension $d$, EMA coefficient $\alpha$). The ratio of gains to increasing complexity raises the question of whether the observed differences reflect a true methodological advancement or simply more fortunate architectural choices on a specific benchmark. Given the magnitude of the gaps, it seems likely that more careful hyperparameter tuning of the simpler baselines (RelGNN or RDL) could close most of the difference, without any of the additional machinery introduced in this work.

---

> ### Author Rebuttal · Authors · 2026-03-31
>
> Thank you for your valuable feedback. Below are our responses to your questions.
>
> 1. **Schema Constraints.** Our aim is to demonstrate that the classical GSL is unable to represent the semantics of given relational instances in a consistent manner. Although the REG learned by classical GSL methods might correspond to some other valid database instances, it may **no longer be consistent with the original schema of the current RDB**. This inconsistency may violate structural constraints, leading to ambiguous or misleading relations. The issue is not just recoverability, but that structural changes can remove or introduce spurious relational information.
> 2. **Deeper Analysis of Table Roles.** **(1) Consistency**: The learned table roles show high consistency, with each table maintaining its role with an average of **87.23%** consistency across 23 tasks (five runs each), and **57.35%** of tables maintaining the same role across all five runs (Response to Reviewer ozyD #3). **(2) Role**: We found that learned table roles **reflect the relative importance of tables**: tables rich in task-relevant features tend to be modeled as nodes, while those with limited features act as edges to facilitate information flow (Response to Reviewer G4rZ #1).  This is consistent with the experimental results. **(3) Interpretability**: Learned table roles **enable interpretable schema graphs**, linking table features to their importance for a task. Examining roles across relations reveals how tables function in information flow, with the **same** table acting **differently** depending on the task context (Response to Reviewer G4rZ #5).
> 3. **Role Selection & GSL.** From the perspective of objectives, the addition and deletion of edges in GSL **essentially modify the information acquisition capability of a GNN during the message-passing process**. Similarly, tables as node/edge also alter the scope of information propagation among nodes, and can thus be viewed as a form of GSL. However, our approach places greater emphasis on preserving the semantic information and the underlying correspondences, mitigating representation collapse and enabling more faithful relational modeling.
> 4. **Hyperparameters.** Although our method has more hyperparameters, in the experiments, the hyperparameters are fixed for 3 tasks (e.g. $\beta=1e^{-6}, \gamma=0.1, d=hid\\_dim/4, \alpha=0.99$ for cls task) because the effective parameter ranges across different datasets are similar, which conveniently demonstrates the generalizability of FROG (see Figure [here](https://anonymous.4open.science/r/FORG-5805/rebuttal/combined.png)). Therefore, **the parameters we used for the search were the same as those of RelGNN** (Appendix F1), thereby also demonstrating the superiority of our method. For the sensitivity analysis of $\beta$ and $\gamma$, please  refer our response to Reviewer ozyD  #4.
> 5. **Full-resolution.** We do not claim that violating full-resolution degrades performance. Rather, our point is that it guarantees on preserving relational semantics. When the correspondence is lost, different relational states may collapse to the same representation. For example, if two users provide similar reviews for different items, and GSL erroneously connects each user to the other item, the users' representations may become indistinguishable, erasing task-relevant distinctions and hindering accurate decision boundaries. Although downstream models need not reconstruct the original data, preserving such structural distinctions can be beneficial for tasks that depend on relational patterns.
> 6. **Hyperedge Implementation.** We decompose hyperedges into standard edges since we only focus on the target node representation. For the target node, each hyperedge connects the same set of entities as its pairwise decomposition, thus preserving information. Moreover, this formulation enables explicit distinction among relation types, which hyperedges alone cannot provide.
> 7. **Search Space of $F_E,F_N$.** We experimented with changing the model's hidden layer dimension. When $hid > hid_0$, the model's performance no longer improves, indicating that the current dimension sufficiently covers the search space. We then focus our search on dimensions where $hid \le hid_0$.
> 8. **Marginal Empirical Improvements.** While the performance gains are sometimes modest, they are consistent across diverse tasks under the same hyperparameter search space as RelGNN (as stated in #4), suggesting a systematic benefit rather than isolated tuning effects. Ablation studies (response to G4rZ #2) further demonstrate the contribution of each component rather than mere tuning, supporting that the proposed method offers a meaningful advancement.
> 9. **Presentation.** We acknowledge that excessive formalism and unclear phrasing may hinder readability, and that the limitations discussion is insufficient; we will improve clarity and include a more thorough limitations section in the revision.

---

> > ### Author Rebuttal · Reviewer_1rQv · 2026-04-01
> >
> > I thank the authors for their response. Most of my core concerns remain unaddressed. I refer to the rebuttal paragraphs as R1-R9 in order of appearance.
> >
> > **Full-resolution (W1, Q1; attempted by R5).** The authors state: "We do not claim that violating full-resolution degrades performance". This rather confirms my original concern. Section 3.3, Propositions 3.3-3.4, Theorem 4.1, and the "full-resolution" framing throughout are all built around this property. If it does not help performance, the theoretical sections are a design motivation, not a principled justification, which is precisely what my review argued. The illustrative example with two users is helpful but informal: it shows what *could* go wrong, not what *does*.
> >
> > **Schema constraints (W1, Q2; attempted by R1).** The rebuttal argues that GSL-modified REGs "may violate structural constraints". I understand the intuition: if each review links to one customer (an FD) and GSL adds a spurious edge to a second customer, the FD is violated. This could be reasonable, but it is never formalized. Propositions 3.3-3.4 only address recoverability, and the informal mention in the text lacks proper analysis. The authors appear to have a valid design intuition: table-level role selection preserves all original FK links, while edge-level GSL can fabricate or destroy connections and violate FDs. But this is presented as a theorem instead of a design principle.
> >
> > **Role selection as GSL (W2, Q3; attempted by R3).** The response argues both approaches "modify the information acquisition capability of a GNN". Under this definition, changing hidden dimensions or activation functions would also qualify as GSL. My question asked for a *precise* connection, not a high-level analogy. Standard GSL modifies edges (a structural change), while FROG makes a per-table binary architectural choice (rather a modeling choice).
> >
> > **Theorem 4.2 assumption (Q4; attempted by R7).** I asked whether the *non-information-preserving assumption* for the intermediate aggregation has been empirically verified. The response discusses embedding dimension sufficiency instead — a separate condition in the theorem. Whether $f_n^1$ is lossy and whether the embedding dimension avoids capacity bottlenecks are different questions. If the aggregation is nearly information-preserving in practice, Theorem 4.2 provides no guarantee.
> >
> > **Hyperparameters (W5, Q5; addressed by R4).** If FROG-specific hyperparameters are fixed per task type and not tuned per dataset, the search budget is comparable to that of RelGNN. I accept this clarification. However, the fixed values had to be determined somehow (presumably via preliminary experiments), and this process is not described in the paper. I would suggest including this detail and the sensitivity analysis in the main text.
> >
> > **Marginal improvements (W5; attempted by R8).** Consistency of small gains does not address the core concern. FROG adds learnable gating, two message passing modules, per-relation GNNs, FD regularization with two losses, and alternating optimization. Improvements are often <1 point within overlapping standard deviations. Each ablated component contributes a small amount, which could equally suggest the full machinery is unnecessary.
> >
> > **Presentation (W2-W4; addressed by R9).** The acknowledgment is appreciated, but this is not merely a phrasing issue. Sections 4.1 and 4.3 contain formalism that adds complexity without proportional insight, as detailed in my original review. Section 4.3 names standard neural network operations as novel "paradigms". Improving clarity would require not just better phrasing but reconsidering which formalism is necessary.
> >
> > **Summary.** The rebuttal does not address my core concerns. R5 confirms the disconnect between theory and practice. R1 and R3 remain informal. R7 answers a different question than Q4. The triviality of Theorems 4.1 and 4.2 (W2, W3) was not discussed at all. R4 is helpful, and I acknowledge it. My assessment remains unchanged.
> >
> > **Suggestions for revision.** The core idea of learning table roles is reasonable and potentially useful for the RDL community. The problem is real, and the empirical exploration (Figure 7, Table 9) is genuinely informative. However, the paper buries this practical insight under unnecessary theory (Theorems 4.1 and 4.2 add no actionable insight), excessive formalism (standard operations dressed as novel paradigms), and over-engineered machinery that yields only marginal gains over simpler baselines. I believe a substantially revised version could become a solid contribution. This would require simpler and more direct language, removing the theoretical sections or reframing them as design motivation rather than necessity, and providing stronger empirical evidence that clearly separates FROG from the baselines. I encourage the authors to undertake such a revision and resubmit to a future venue.

---

> > > ### Author Response · Authors · 2026-04-03
> > >
> > > We sincerely thank the reviewer for the detailed follow-up comments and continued engagement. The following is a supplement to our response.
> > >
> > > - **Supplement of R5 & 1**: We clarify that our intention is not to position full-resolution as a performance guarantee, but as a principled perspective on relational representation design. In relational settings, structural information defines semantics, and our formulation aims to distinguish structure-preserving transformations from those that may alter or distort such semantics. We agree that these results are better interpreted as design principles rather than strict theoretical necessity, and we will revise the paper accordingly.
> > >
> > > - **Supplement of R3**: We agree that our formulation differs from classical edge-level GSL. However, table role assignment does induce concrete structural changes in the resulting graph. Specifically, representing a table as an edge instead of a node reduces the number of nodes (|T|->|T|-1), reduces the number of edges(|rel|->|rel|-1), and  increases edge feature richness. These changes directly modify the graph structure.
> > >
> > > - **Supplement of R7**: We would like to clarify that this response addresses whether the encoding dimension is “sufficiently large” to support a function family capable of covering $F_E,F_N$ (W3.2). That's why we explore how the size of the hidden layer affects performance. For  non-information-preserving assumption, most aggregation functions (such as sum, mean, etc.) are many-to-one mappings, which means that they are not invertible. This causes loss of information due to different neighborhood configurations being mapped to the same representation.
> > >
> > > - **Supplement of R8**: Statistically, in Tables 1 and 2, there are only around three cases where the optimal and suboptimal results overlap, and in some of these cases, the overlapping regions are quite small. For the regression tasks in Table 3, we acknowledge that the performance differences are less pronounced compared to those in Tables 1 and 2 (due to the relatively concentrated distribution of the target values). Nevertheless, considering the results across all three tables, there are still notable performance improvements overall, which demonstrates the effectiveness of the proposed method under structural constraints.

---

### Official Review · Reviewer_xUwP · 2026-03-13

**Soundness:** 4
**Presentation:** 3
**Significance:** 3
**Originality:** 4
**Overall Recommendation:** 5
**Confidence:** 5

**Summary:**

This paper introduces FROG that introduces graph structure learning into relational deep learning for the first time, as far as I known. FROG learns to map relational tables either as nodes or as edges in a schema graph while preserving the full-resolution property. It also designs relation-driven message passing and a functional dependency loss to respect database semantics. Experiments on benchmarks show consistent improvements over existing RDL methods.

**Compliance With Llm Reviewing Policy:**

Affirmed.

**Key Questions For Authors:**

1. In Equation (6), the gate $\tilde{g}_{v,r}^{(l)}$ is computed using both node‑level and edge‑level representations. Are these representations obtained from separate GNN passes? If so, does this double the computation per layer?

2. The FD loss encourages embeddings of entities linked by an FD to lie in a low‑rank subspace. However, some FDs may be approximate or noisy in real data. Does FROG assume strict FDs? How would it behave if the database contains violations?

**Limitations:**

Yes.

**Strengths And Weaknesses:**

# Strengths
1. Existing RDL methods predefine the schema graph, and GSL techniques lose information on RDBs. FROG is the first to enable end-to-end schema graph optimization with full information preservation.

2. The authors formalize full resolution from an information‑theoretic perspective and prove that both table‑as‑node and table‑as‑edge mappings satisfy it. Theorem 4.2 provides an intuitive information‑theoretic justification for preferring edge‑level modeling in certain cases.

3. The paper does not stop at only performance. It analyzes learned roles (Section 5.5) and visualizes how FD loss structures the embedding space (Figure 8). The appendix contains detailed proofs, dataset descriptions, and additional experiments, making the work reproducible.

# Weaknesses
1. There is a lack of specific analysis of the time cost. In the data preprocessing of the paper, obtaining the relationship of 2-hop neighbors is required, which may result in significant time consumption.

2. Some sections are difficult to follow (e.g., the table-role optimization mechanism). More intuitive explanations or examples could enhance readability.

3. While full resolution is theoretically appealing, in practice one might tolerate minor information loss if it leads to significant efficiency gains. The paper does not discuss whether full resolution is always necessary or whether there are trade‑offs.

---

> ### Author Rebuttal · Authors · 2026-03-31
>
> Thank you for your valuable feedback. Below are our responses to your questions.
>
> 1. **Time Complexity.** As shown in Algorithm 2, the time complexity of 2-hop processing is $O(|\mathrm{rel_E}|\, |E| \log |E|)$. We do not enumerate all valid 2-hop samples to avoid higher complexity and inference overhead. With broadcasting, FROG introduces a moderate computational overhead compared to RelGNN, from data preprocessing to inference, as shown below. We agree that your concern regarding time complexity is well-founded. In future work, we will consider precomputing global 2-hop relations prior to training, which can reduce inference time by approximately 20%.
>
>    | Time (s/batch)    | RelGNN | FROG (2 hop processing ) | FROG (inference) |
>    | ----------------- | ------ | ------------------------ | ---------------- |
>    | user-ignore       | 0.04   | 0.01                     | 0.06             |
>    | driver-dnf        | 0.07   | 0.03                     | 0.17             |
>    | post-post-related | 0.08   | 0.02                     | 0.09             |
>
> 2. **Computational Overhead.** The FROG only perform incremental computation in the edge convolution within the GNN part of the whole model. The related components are edge convolution and gate, which constitute the incremental part compared to other methods. Their size grows linearly with the number of roles in table-as-edge. The parameter increments for some datasets are shown below. We believe that this increase in the number of parameters is acceptable given the improvement in performance.
>
>    |           | Total Para | Edge Conv+Gate Para | Proportion |
>    | --------- | ---------- | ------------------- | ---------- |
>    | rel-hm    | 2,138,499  | 74,112+16,642       | 4.2%       |
>    | rel-trial | 8,089,230  | 280,322+91,531      | 4.6%       |
>    | rel-event | 6,805,643  | 658,436+198,150     | 12.6%      |
>
> 3. **FD Loss.** FROG does not assume that the learned representations will strictly satisfy FDs; instead, FDs are incorporated as a regularization signal to promote consistency with the underlying RDB. At the schema level, the Foreign-Primary Key in an RDB are designed to satisfy FD properties by definition (as required by the RDB itself). If violations occur in the actual data, the FD normalization mechanism serves as a soft constraint.
>
> 4. **Full Resolution.** We consider full-resolution as the foundation for **preserving relational semantics**. Unlike ordinary graphs, the normalized structure of relational databases means that a minor structure loss can lead to semantic confusion. FROG addresses the trade-off between semantic preservation and model performance through “table role modeling”: while maintaining FR, the model dynamically optimizes message-passing paths. Additionally, introducing functional dependency constraints leverages prior knowledge to reduce the search space, improving learning performance while ensuring semantic reachability.
>
> 5. **Clarity of Exposition.** We acknowledge that some complex symbolic expressions and the lack of intuitive explanations may make it difficult for readers to understand our ideas. We thank the reviewer for the valuable comments, and we will make appropriate revisions to improve readability.

---

> > ### Author Rebuttal · Reviewer_xUwP · 2026-04-04
> >
> > Thanks for the response. My main concerns have been well addressed.

---

### Decision · Program_Chairs · 2026-04-30

**Decision:**

Accept (regular)

**Comment:**

This paper proposes a relational deep learning framework that enables full-resolution graph structure learning via learnable table roles.It addresses an important problem in RDL that moving beyond fixed schema graphs. This perspective could have meaningful impact on the community and multiple reviewers highlighted the novelty and originality of the work. While one reviewer raised concerns regarding the theoretical framing, these issues do not undermine the technical soundness of the method, and the authors have provided responses to address the questions. Overall, the paper demonstrates sufficient innovation, solid methodology, and is recommended for acceptance.